# Emergent color categorization in a neural network trained for object recognition

Jelmer P de Vries[1]*, Arash Akbarinia[1], Alban Flachot[1,2], Karl R Gegenfurtner[1]*

[1]Experimental Psychology, Giessen University, Giessen, Germany; [2]Center for Vision Research, Department of Psychology, York University, Toronto, Canada

**Abstract** Color is a prime example of categorical perception, yet it is unclear why and how color categories emerge. On the one hand, prelinguistic infants and several animals treat color categorically. On the other hand, recent modeling endeavors have successfully utilized communicative concepts as the driving force for color categories. Rather than modeling categories directly, we investigate the potential emergence of color categories as a result of acquiring visual skills. Specifically, we asked whether color is represented categorically in a convolutional neural network (CNN) trained to recognize objects in natural images. We systematically trained new output layers to the CNN for a color classification task and, probing novel colors, found borders that are largely invariant to the training colors. The border locations were confirmed using an evolutionary algorithm that relies on the principle of categorical perception. A psychophysical experiment on human observers, analogous to our primary CNN experiment, shows that the borders agree to a large degree with human category boundaries. These results provide evidence that the development of basic visual skills can contribute to the emergence of a categorical representation of color.

## Editor's evaluation

This paper addresses the long-standing problem of color categorization and the forces that bring it about, which can be potentially interesting to researchers in cognition, visual neuroscience, society, and culture. In particular, the authors show that as a "model organism", a Convolutional Neural Network (CNN) trained with the human-labelled image dataset ImageNet for object recognition can represent color categories. The finding reveals important features of deep neural networks in color processing and can also guide future theoretical and empirical work in high-level color vision.

*For correspondence:
vriesdejelmer@gmail.com (JPdV);
gegenfurtner@uni-giessen.de
(KRG)

**Competing interest:** The authors declare that no competing interests exist.

## Introduction

Color vision is a prime example of categorical perception, and, as such, has received considerable attention across several research domains (*Harnad, 1987*). Being dependent on both linguistic and perceptual processing has made it difficult to pinpoint the mechanisms responsible for the emergence of color categories, and determine why particular colors are grouped the way they are. This has led to a protracted debate as to what extent categorization develops *universally* (independent of local language and culture) and to what extent it is *relative* to local communication (for an elaborate discussion on the Sapir-Whorf hypothesis see *Kay, 2015*; *Kay and Kempton, 1984*). Proponents of the universalist view have pointed toward the overlap in focal colors across different cultures (*Regier et al., 2005*). Furthermore, it appears that categories can to a certain degree emerge independent of language development: Pre-linguistic infants pay more attention to color changes crossing categorical borders than color changes within categories (*Skelton et al., 2017*) and several animal species respond to color categorically (*Caves et al., 2018*; *Jones et al., 2001*; *Poralla and Neumeyer, 2006*). Relativists (*Davidoff, 2001*), however, point toward the difficulty children have acquiring color names

(*Roberson et al., 2004*) and the case of a patient whose language impairments were associated with color sorting problems (*Roberson et al., 1999*). Also, while categorization has been found in several animals, researchers have failed to find categorization in some primates (e.g. baboons) and it has been argued that the methodology of other primate studies was biased toward finding categorical results (*Fagot et al., 2006*). Moreover, just as universalists point to the strong commonality among the development of color categories, proponents of the relativist view highlight differences (*Roberson et al., 2000*; *Roberson et al., 2005*).

With the apparent contradictions in findings, in more recent years, the universalist versus relativist debate evolved from contrasting two extremes, to looking at how different factors contribute to the process of categorization (*Kay and Regier, 2006*; *Steels and Belpaeme, 2005*). Importantly, recent advances have also started taking into account the varying utility of colors (*Conway et al., 2020*; *Gibson et al., 2017*; *Zaslavsky et al., 2019*). One seminal study shows that warm colors are communicated more efficiently than cool colors (overall) and, importantly, that at the cultural level categorical differences between languages are the result of differences in *usefulness of color* (*Gibson et al., 2017*). Subsequent papers have demonstrated that utilizing a perceptually uniform color space in combination with concepts from communication theory, such as an information bottleneck (*Zaslavsky et al., 2018*) or rate distortion (*Twomey et al., 2021*) can be powerful in modeling the shape of color categories. Notably, a recent study, where communicating deep neural networks played a discrimination game, demonstrated that allowing continuous message passing made the emergent system more complex and decreased its efficiency (*Chaabouni et al., 2021*).

While the modeling approaches incorporating communication principles have proven powerful in predicting categorization characteristics, the strong reliance on communication does not address the existence of what appears to be categorical behavior for color in the above-mentioned pre-linguistic infants and various animals. Also, a recent case study shows that color *naming* can be impaired while color *categorization* remains intact, emphasizing that in humans the link between communication and categorization can be decoupled (*Siuda-Krzywicka et al., 2019*). Furthermore, the trend of looking at the ecological relevance of color in shaping categories has also extended to animal research. Host birds rely on a single-threshold decision rule in rejecting parasite eggs rather than the dissimilarity in color to their own eggs (*Hanley et al., 2017*) and female zebra finches categorically perceive the orange to red spectrum of male beak color (*Caves et al., 2018*). These latter findings, particularly, emphasize that in general the utility of color is at the basis of color categories. Because the utility of color will also be reflected in communication, the fact that communicative concepts are powerful in modeling the shape of categories does not necessarily prove a causal relationship. Moreover, in the case study by Siuda-Krzywicka and colleagues their patient, RDS, tries to use objects to link names to colors ("this is the color of blood; it must be red" page 2473). A link between color categories and objects would be able to bridge the discrepancy between models that rely on communicative concepts to incorporate the varying usefulness of color and the experimental findings on pre-linguistic categorical effects. If object recognition is a driver of color categories, this would be a crucial step towards an answer to *why* and *how* color categories spread across perceptual color space the way they do. The possible causes range from visual perception being shaped based on the physical properties of the world all the way to the need for efficient communication about color.

Here, we focus on color categorization as a potential emergent property of acquiring object recognition. Specifically, we investigate whether a categorical representation of color emerges in a Convolutional Neural Network (CNN) trained to perform an object recognition task on natural images. With this, our approach is not to model specific color category data directly, nor to model specific brain processes. Rather, we investigate whether a categorical representation of color can emerge as a side effect of acquiring a basic visual task; object recognition. The CNN is an excellent candidate for this purpose as unlike in any living species we can control its visual diet and train it on one specific task. Previously, many studies on the representation of color in a CNN rely on physiological style approaches (*Engilberge et al., 2017*; *Flachot et al., 2020*; *Flachot and Gegenfurtner, 2018*; *Rafegas and Vanrell, 2018*). Considering color categorization is likely a higher order process, we rely on classical principles from a long history of psychophysical studies to study the emergence of color categorization. In the CNN this translates to replacing the final layer with a new classifier which can be trained to perform alternate tasks on the representation the network builds (over the layers) to perform object

classification on. Our findings show that a CNN trained for object recognition develops a categorical representation of color at the level where objects are represented for classification.

## Results

### Border invariance

Perceptual research in non-human species requires indirect measures: in numerous species *match-to-sample* tasks have successfully been utilized for a long time to study visual perception (*Kastak and Schusterman, 1994*; *Skinner, 1950*). Pigeons, for example, were trained to match colors to one of three main color samples (*Wright and Cumming, 1971*). Subsequently, novel colors were introduced to evaluate to which of the sample colors they were matched to. This allowed the authors to find the hues where pigeons switch from one color to another. When they repeated the experiment with different training colors, they found crossover points to be similar across experiments, indicating a categorical perception of color. Here we use a similar approach to evaluate the color representation of a ResNet-18 CNN (*He et al., 2016*) that has been trained on the ImageNet dataset, where objects in natural images have to be classified (*Deng et al., 2009*). First, we replaced the original output layer (which classified the 1000 objects) with a new classifier. The new classifier was then trained (while keeping the original weights in the rest of the network constant) to classify stimuli containing a single word of a specific color (selected from narrow bands in the HSV hue spectrum at maximum brightness and saturation). For stimulus examples see *Figure 1A*; the hue spectrum with corresponding training bands is found in *Figure 1B*. In a second step we evaluate the retrained network using colors from the *whole* hue spectrum and determine to which classes these colors are generalized. As shown in *Figure 1C*, colors from outside of the training bands are largely classified to the neighboring bands in a consistent manner. As a consequence, the best color match for each point on the hue spectrum can be determined in a straightforward manner by taking the mode (see *Figure 1D*). The important question regarding categorization now is what determines the transitions (or borders) between the matches? One option is that the borders are dependent on the positions of the training bands, meaning a shift in these bands should translate to a shift in borders. Alternatively, the borders between the colors could be due to a categorical representation of color in the network. In this latter case we expect the borders to be (at least partially) invariant to shifts in the training bands. To investigate this, we repeated the above process many times over, while slightly shifting the training bands over the iterations. Unfortunately, there is no prior work on color categories in CNNs to base the number bands on. From the human perspective 6 of the 11 basic color terms appear in the spectrum (red, orange, yellow, green, blue, and purple), as such we varied the number of output classes from 4 through 9 in an exploratory manner. The result for 6 output classes has been visualized in *Figure 1E*: As the training bands are gradually shifted (indicated by the black lines) we observe that the borders between categories appear largely invariant to these shifts in training bands.

To determine whether the borders are consistent across the shifting training bands, we plot the *transition count* that indicates the co-occurrence of borders in *Figure 2A* (i.e. the number of times a border occurs in a specific location, while the bands are shifted). Note that while *Figure 1E* only displays the results for a network trained with 6 output nodes, the full result depends on borders found when training the network with 4 through 9 output nodes. As those results are collapsed, inspecting the transition count shows that there are about 7–8 discontinuities in the hue circle. Using a basic peak detection algorithm that relies on neighboring data points to find peaks, 7 peaks are found (red dots in *Figure 2A*). Utilizing those peaks, we divide the hue spectrum in 7 different regions (*Figure 2C*), and averaging the colors in each region (weighing them based on the reciprocal of the transition count) results in 7 colors that can be seen as representing each category.

Since the degree of invariance is not constant over the different borders, the question becomes what we consider sufficient evidence for finding categories and how we can exclude the possibility that this is a chance finding. To ensure the latter is not the case, we have rerun the same experiment using the same and similar architectures (notably a ResNet-34, ResNet-50 and ResNet-101; see Appendix 1: Network Repetitions). In *Appendix 1—figure 1* one can see, that given a similar architecture, similar border locations are obtained. In Appendix 8 we have also included a number of different architectures to ensure the current results are not exclusive to ResNets. Quantifying how 'categorical' the current results are is more complicated, however. If the color representation is strictly categorical, the

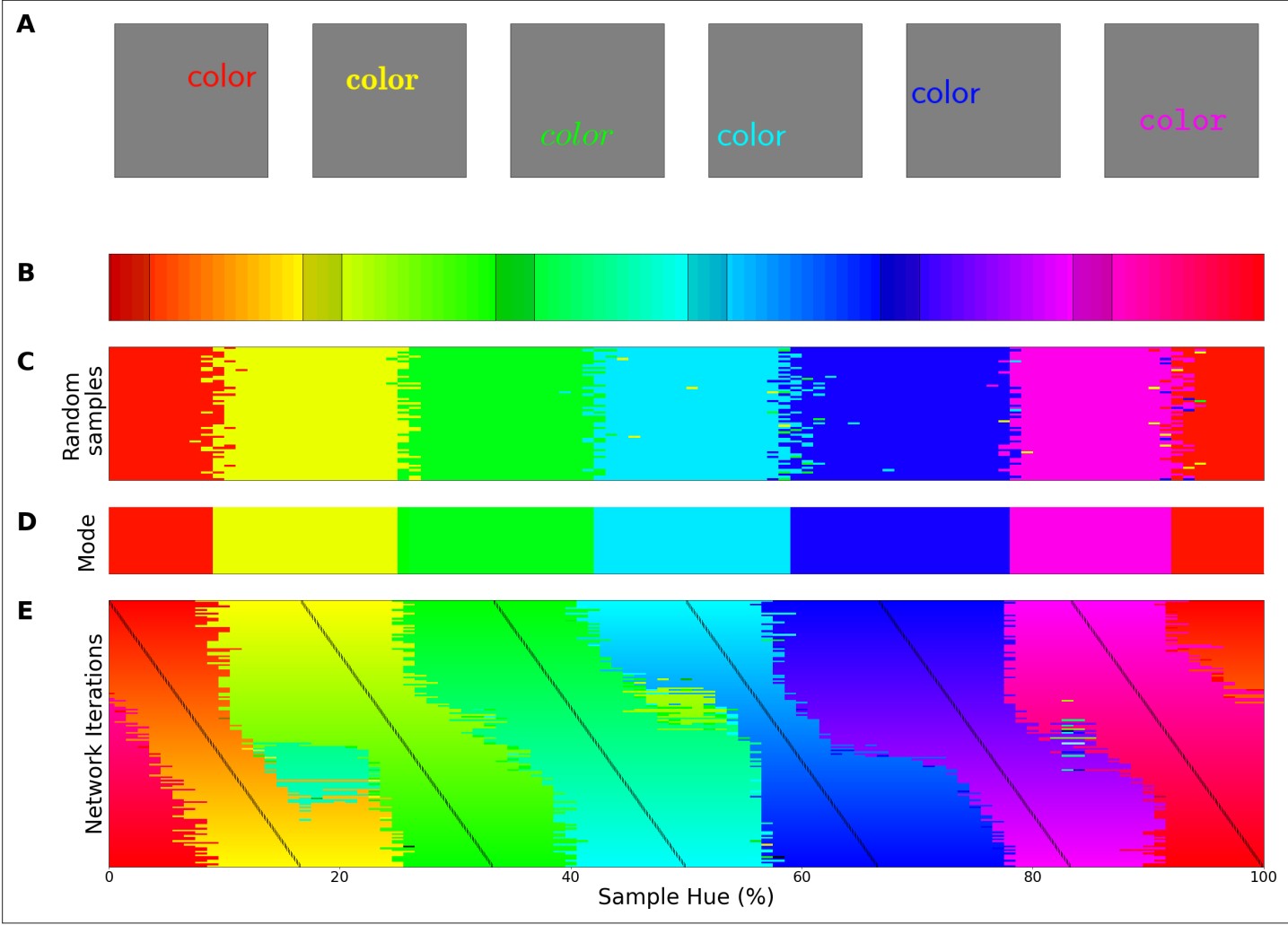

**Figure 1.** Invariant border experiment. (**A**) Six stimulus samples corresponding to the primary and secondary colors in the hue spectrum (red, green, blue, yellow, cyan and magenta, respectively). (**B**) Hue spectrum from HSV color space (at maximum brightness and saturation). The colors for each class are selected from narrow, uniformly distributed, bands over the hue spectrum. Bands are indicated by the transparent rectangles. (**C**) Evaluation from the training instance for which the bands are depicted in **B**. In each instance, the same ImageNet-trained ResNet-18 is used, but a novel classifier is trained to perform the color classification task with the number of output nodes corresponding to the number of training bands. Each individual pixel represents one classified sample, colored for the class it has been assigned to (i.e. using the hue of the center of the training band). (**D**) A one-dimensional color signal produced by taking the mode of each pixel column in **C**. In this manner, we obtain the overall prediction for each point on the spectrum and can determine where the borders between classes occur. (**E**) Results of all instances trained on 6 bands as they are shifted through the hue spectrum. Each row represents the classification of a single network (as in **D**), trained on 6 bands, the center of which is marked by a black tick (appearing as black diagonal lines throughout the image).

result in *Figure 1E* would primarily show straight lines. On the other hand, if the network incorporated some form of a continuous representation of color, we expect borders to shift with the shifting color bands. In the latter case diagonal transitions, following the diagonal of the training bands, should be most prominent. In the categorical case, the highest cross-correlation between individual rows would be found by keeping them in place. However, in the continuous case, the maximum cross-correlation would be found by shifting the rows relative to each other. We can contrast these two cases by calculating, for pairs of rows, the lateral shift that produces the highest cross-correlation and inspect the frequency for each of the found shifts. If the borders move along with the training colors, the shifts should be distributed more uniformly (−7 to 7; range is equal to the width of 7 uniform bands). If the borders are stable across the different training iterations, we expect most shifts to be small (say between −2 to 2). Simulations for the categorical and continuous case are shown by the green and purple curve in *Figure 2E*, respectively (see Appendix 2: Simulation Classification for more details).

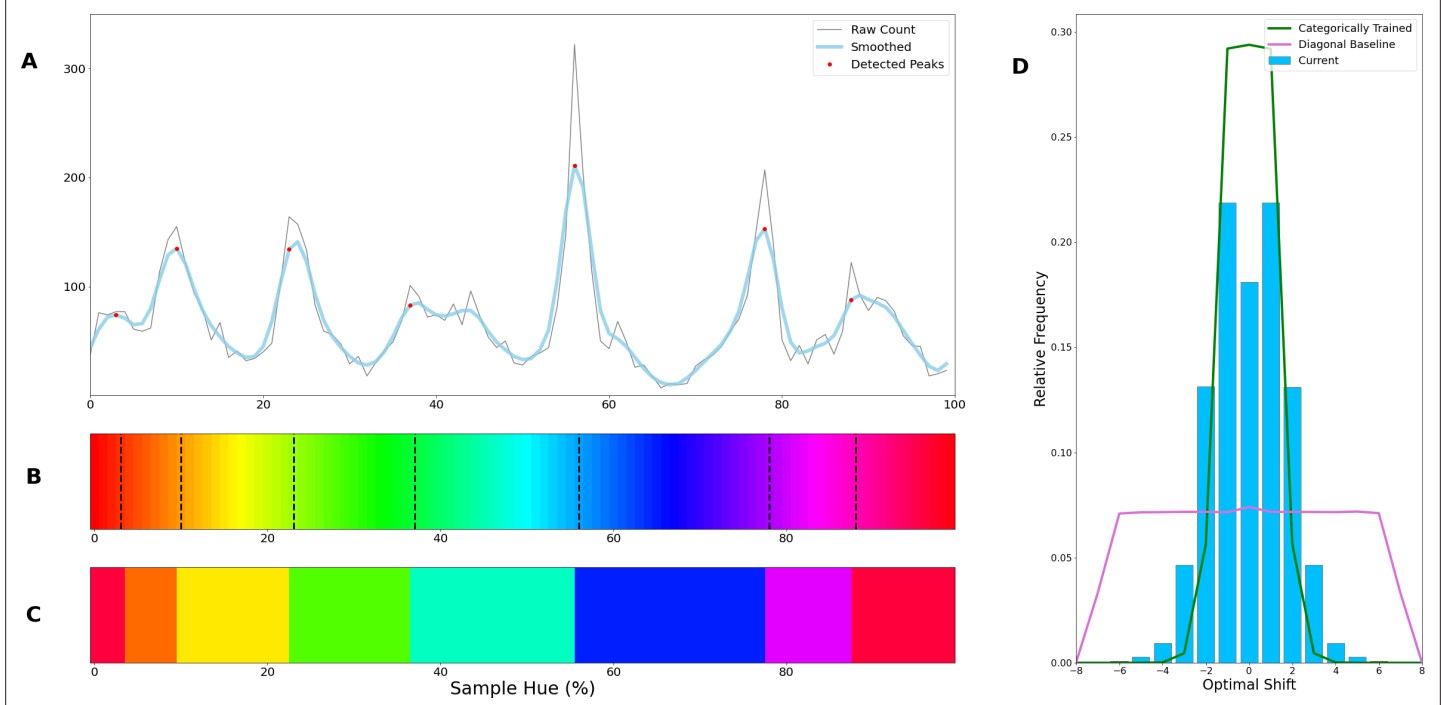

**Figure 2.** Border transitions in the color classifications. (**A**) Summation of border transitions, calculated by counting the border transitions (as depicted in *Figure 1E*) for each point on the HSV hue spectrum (thin grey line). A smoothed signal (using a Gaussian kernel; blue thick line) is plotted to reduce the noise. Peaks in the signal (raw count) are found using a simple peak detection algorithm (findpeaks from the scipy.signal library) and indicated in red. (**B**) The peaks are superimposed on the hue spectrum as vertical black dotted lines. (**C**) Category prototypes for each color class obtained by averaging the color in between the two borders (using reciprocal weighing of the raw transition count in **A**). (**D**) For each row (as in *Figure 1E*), the optimal cross-correlation is found by comparing the row to all other rows in the figure and shifting it to obtain the maximum correlation. In blue we plot the distribution of shifts when 7 output classes are used (as we appear to find 7 categories). For comparison, we plot the result of a borderless situation (where borders shift with training bands) in purple and in green the result for a network trained from scratch on the found 7 color categories.

The histogram plotted in light blue shows the actual data, which more closely follow the green line, representing the categorical simulation. A Fisher's Exact Test shows the difference between the count distribution is significant for all comparisons (p<0.0001; both pairwise and three-way). While the distribution in *Figure 2D* relies on comparing each individual row to all rows, this leads to the inclusion of doubles. For the statistical test, we therefore only include each comparison once and take the shift as a positive distance value. The counts for each positive shift occurrence were entered into the test. Even though all comparisons were significant, it is important to note that the overlap between the current data and the shifting simulation is much smaller than with the categorical simulation (48% vs 74%). As such, despite a significant difference for both comparisons, it seems that the current findings more closely match with the categorical simulation. The significant difference between these two can be explained by the fact that directly training the color categories will lead to a less noisy color representation than when a network is trained on objects. Also, comparing the classification from the simulated data in *Appendix 2—figure 1* (Appendix 2: Simulation Classification) we can see that the color classifications from the current ResNet-18 and the categorically trained ResNet-18 are very similar and both deviate considerably from the continuous simulation.

Our procedure requires the use of a hue spectrum that wraps around the color space while including many of the highly saturated colors that are typical prototypes for human color categories. We have elected to use the hue spectrum from the HSV color space at full saturation and brightness, which is represented by the edges of the RGB color cube. As this is the space in which our network was trained, minimal deformations are introduced. Other potential choices of color space either include strong non-linear transformations that stretch and compress certain parts of the RGB cube, or exclude a large portion of the RGB gamut (yellow in particular). Varying the hue in the HSV space, however, does not only change the color of the word stimuli, but also their luminance. To ensure the current borders do not stem from a spurious correlation between color and luminance, we have rerun the

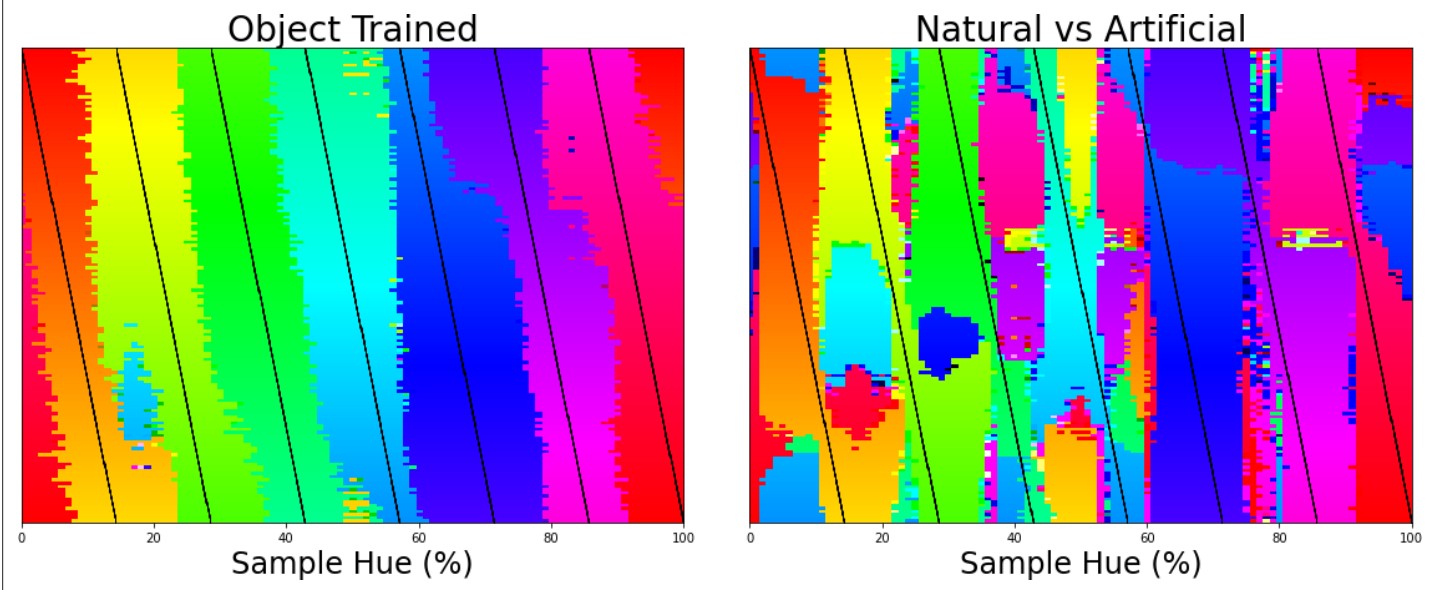

**Figure 3.** Left the classification of colors for 7 training bands being shifted over the hue spectrum as in *Figure 1E*. Right the same analysis, but applied to a network trained to classify scenes (natural vs. artificial).

current experiment with different stimuli that include luminance distractors and a variable background luminance (see Appendix 3: Luminance Variation). While border locations are not a perfect one-to-one match with the current results, they are similar and a categorical representation is again found. With the introduction of distractors and variation in background luminance the network can rely only on kernels coding purely for color and not a combination of color and luminance to perform the task. We should note that the analysis does not suggest that the representation of color is completely luminance independent. Some color categories are inherently dependent on luminance contrast. For instance, the only difference between orange and brown is relative luminance (see for instance: Figure 1B in *Lindsey and Brown, 2009*). The current analysis only excludes the notion that the borders are solely based on luminance. We also explored an alternative hue spectrum from a single plane of RGB color space. This led again to a categorical representation, albeit more noisy, presumably due to the reduction in chromatic contrast (see Appendix 4: Circular Color Spectrum).

The final important question that arises from the current results is to what extent they are related to the acquisition of object classification, and to what extent the architecture of our model and the colors in the dataset drive the categorical representation. Previous studies have already shown that the perceptual structure can in part explain color naming (e.g., *Zaslavsky et al., 2019*). However, what we believe makes the current finding particularly interesting is that the current results stem from a network trained for object recognition and, importantly, we find the categorical color representation in the final layer of the network (the network's representation that enables object classification). This suggests that this categorical color representation is beneficial to the object classification task. If this color representation is indeed present in the final layer to the benefit of object classification, there should also exist other classification tasks which do not result in a categorical representation of color in the CNN. Therefore, we trained the ResNet-18 architecture on the same images, but on a different classification task, specifically, on distinguishing natural from man-made scenes. We keep the color distribution of the input the same by using the same ImageNet dataset for this task. The color classifications of the networks trained on different tasks can be found in *Figure 3*, with the left panel showing the classification of colors in the original object-trained ResNet-18 and the right panel showing the classification of the ResNet-18 trained to distinguish the two scene types. The color representation for the latter ResNet-18 is not one that dissects the hue spectrum in human-like categories. As we do again observe a number of straight transitions, one could argue that the color representation does show categorical traits, however, it is clear that the color representation does not follow the hue spectrum in a manner similar to that of the object-trained version. As such, it appears that the representation build for object recognition is considerably closer to what we expect from humans.

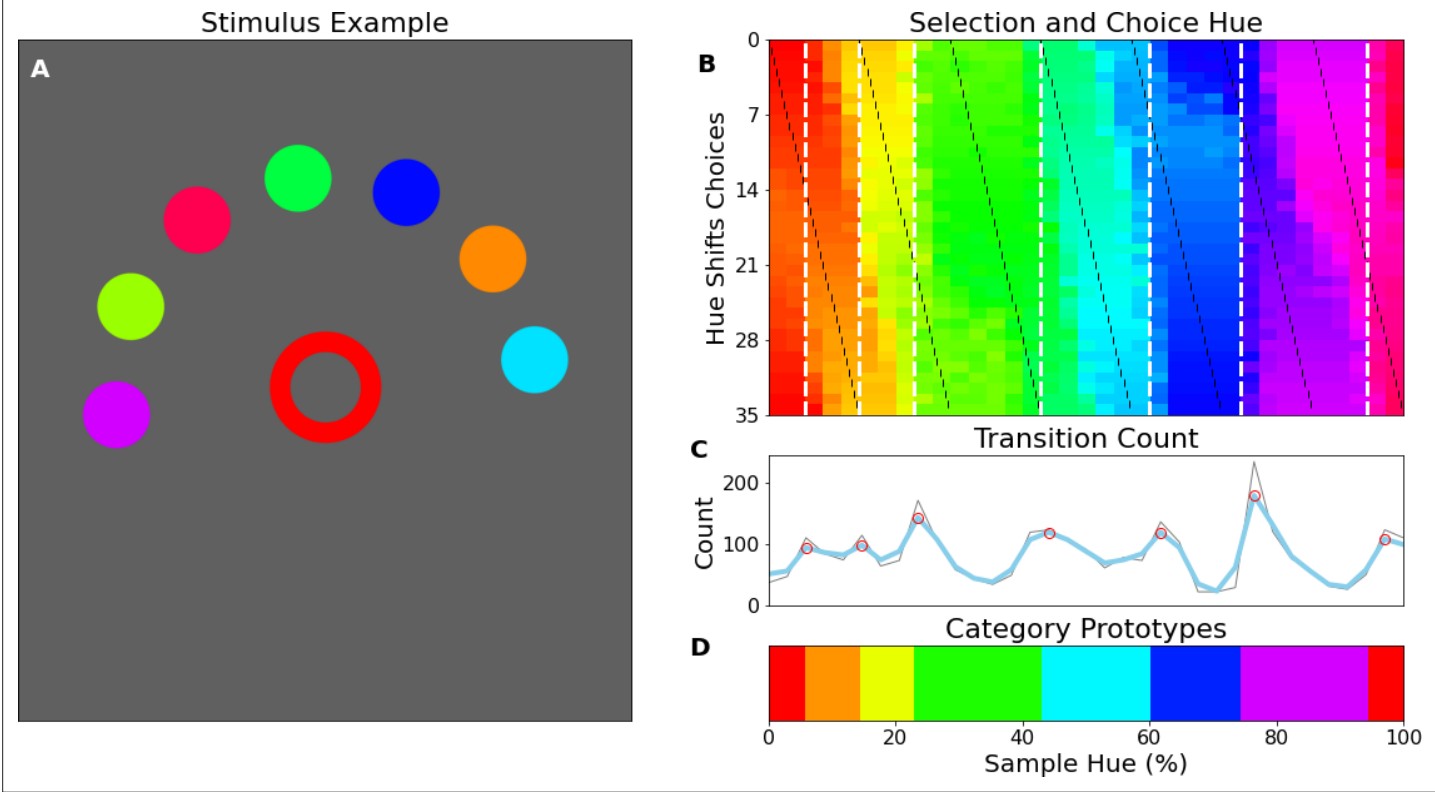

**Figure 4.** Human psychophysics. (**A**) Example display of an iPad trial. The observer's fingertip is placed in the central circle (white at the start of the trial) upon which it shrinks and disappears (over 150ms); subsequently it reappears in the target color (red in the current example). In the current display, the peripheral choices, on the imaginary half-circle are rotated slightly counter-clockwise, this alternates every trial between clockwise and counter-clockwise. (**B**) Color selections of all observers averaged over hue. The target colors are represented on the x-axis and each row represents a different set of peripheral choices, the hues of which are indicated by the black tick marks. Each pixel in the graph is colored for the averaged observers' choice. The white vertical dotted lines indicate the estimated categorical borders based on the transition count. (**C**) The transition count as in *Figure 2A*, but now cumulated over observers, rather than network repetitions. (**D**) As in *Figure 2C* we have determined the prototypical color of each category by calculating the average, weighted by the reciprocal of the transition count.

## Human psychophysics

The previous experiment shows that the network has built a representation of color that appears categorical in nature. However, the approach and analysis do not have a direct equivalent in human color categorization studies. To evaluate whether the results from this novel approach are akin to human behavior, we measured human observers on a match-to-sample task modeled after the task for the CNN. Ten human observers performed an experiment where (centrally) presented color samples had to be matched to one of 7 peripheral colored disks (see *Figure 4A* for an example stimulus display). As in the CNN experiment, to determine borders we vary both the central target color, as well the hues of the 7 peripheral choices the observer has to pick from over trials.

To allow for comparison with the previous experiment, we plot the results in a similar manner. The human data can be found in *Figure 4B–D*; In *Figure 4B*, we see that much in the same way as in the previous experiment there are clear locations in the hue spectrum where borders occur regardless of the hues of the peripheral disks. Particularly, as with the neural network we find categories are narrower in the red-yellow range, while the green region is the broadest. The transition count in *Figure 4C* highlights that, much in the same way as with the CNN experiment, the borders are not absolute and some borders are stronger than others.

However, we should also acknowledge that considering the repetitions here are different observers, the variance is partly driven by different perceptual experiences and potentially also by the application of different strategies. Still, the variance is also present in individual observers and observing *Figure 4D* where we have estimated the color prototypes for each human color category, shows a pattern that is highly analogous to the CNNs prototypes. To ensure the border count in *Figure 4C* is

reliable, we tested for significance by comparing the correlation among observers to a bootstrapped null-distribution (see methods section) and find the correlation of the transition count is significantly different to transition counts of bootstrapped observers for whom counts do not align ($p < 1 \times 10^{-6}$).

## Evolutionary algorithm using principles of categorical perception

The generalization over neighboring colors and the invariant discontinuities between them are consistent with the notion that the network builds a categorical representation of color. Nevertheless, the lack of a broad understanding of CNNs in general and particularly their representation of color makes us wary to draw a definitive conclusion from only one type of analysis. To evaluate whether the colors within the boundaries can indeed best be seen as belonging to categories, we turn to the concept of *categorical perception*, where differences between colors within the same category are seen as smaller than differences between colors from different categories (*Goldstone and Hendrickson, 2010*). If the discontinuities we found indeed mark borders as in a categorical representation of color, we expect that generalizing colors falling between two borders should be *easier* than generalizing colors that cross discontinuities. In humans, categorical perception is often studied using reaction time tasks (*Kay et al., 2009*; *Winawer et al., 2007*; *Witzel and Gegenfurtner, 2011*), but a direct analogue of reaction times in neural networks is not available. There is however a different temporal performance measure available: the ability to evaluate how quickly a network can learn a task. If the boundaries found in the Invariant Border Experiment indeed resemble categorical borders, it should be faster for the network to learn to generalize colors within two neighboring discontinuities to a specific class than to generalize colors crossing discontinuities.

Specifically, in the current experiment, we evaluate how well a set of borders fits the color representation of the network by evaluating how easily sets of 2 narrow training bands placed directly inside of each neighboring border pair can be generalized to single classes. One straightforward way of evaluating the discontinuities found above would be to make direct comparisons based on their locations. However, this would mean either using biased comparisons based on the borders found above, or require an almost infinite number of comparisons to ensure we compare the found borders to every alternative. To avoid this, rather than using the previously found borders as a starting point, we developed a search algorithm that uses the principle of categorical perception to find the optimal

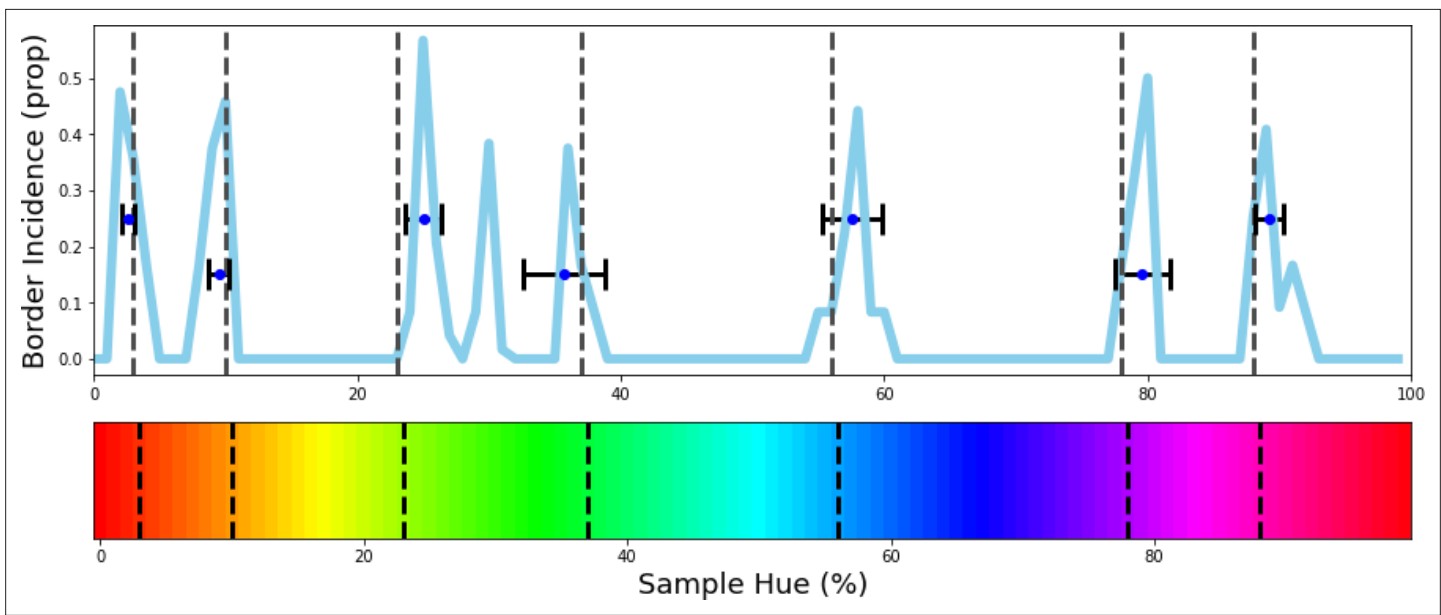

**Figure 5.** Evolutionary results. The evolutionary algorithm is repeated 12 times and we calculate the frequency of borders in the top 10 border sets of each repetition. The resulting frequencies are plotted in blue. Border-location estimates from the Invariant Border Experiment are plotted in the graph and on the hue spectrum in dotted black vertical lines for comparison. Running the algorithm 12 times results in 120 solutions with 7 borders each. The 7 borders are ordered from left to right and then, from the 120 solutions we take the median border for the 1st through 7th border. We have plotted these medians as points, including a horizontal errorbar that indicates a standard deviation to visualize the variability of these values. As can be seen, the estimate for each column closely agrees to the estimate from the invariant border experiment.

set of borders for the network from scratch. The only information taken into account from the previous experiment is that approximately 7 borders were found: An evolutionary algorithm is initialized with 100 sets of 7 randomly placed borders. Allowing the network to train for a limited number of epochs (the number of times all images in the dataset are passed through the network) we evaluate the *performance* of each of the randomly initialized border-sets. In evolutionary algorithms the fitness function serves to evaluate how well solutions perform. As such the short training of the network serves as the fitness function for the border-sets. Using the fitness of each set, the best 10 performers are copied to the next generation (elitism) as well as a set of 90 novel border-sets. The latter sets are generated by randomly selecting parents from the previous generation (with a bias for better performing ones) and recombining their borders to create new border sets. By allowing this process to run for some 40 generations, it converges to a specific set of borders with little variation between the top ten performers. As evolutionary algorithms are not guaranteed to converge to a global optimum, we ran the algorithm 12 times to ensure the results are consistent. In the current case, the borders should be less variable, but still line up with the peaks of the invariant border experiment.

*Figure 5* shows where the evolutionary algorithm places the borders and we see a strong correspondence with the previously found borders (indicated by black vertical dotted lines). The only exception appears in the border between green and turquoise. However, note that this is also the region of color space where the transition count (as shown in *Figure 2A*) did not show a sharp peak, as was the case for most other borders. Interestingly, this is also the border where the largest variability is observed for human observers (*Hansen and Gegenfurtner, 2017*). The current results suggest that the best explanation for the discontinuities we observed in the Invariant Border Experiment are that they indeed represent categorical borders dividing the color space. The fact that color categorization emerges in a CNN trained for object recognition underscores that color categorization may be

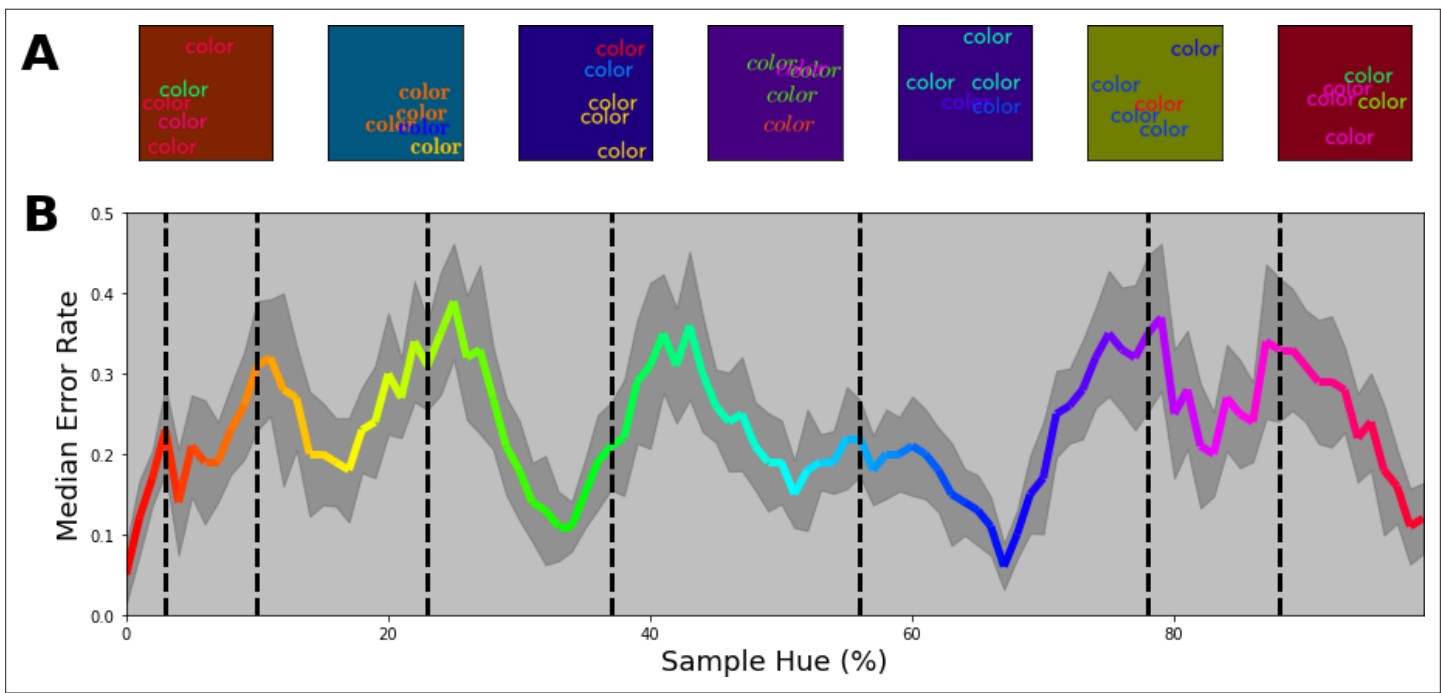

**Figure 6.** Multi-colored stimuli classification performance. (**A**) 7 example stimuli, each sampled from a different color band. Each stimulus consists of three equally colored (target) words of which the color is determined by the selected class. Subsequently, two randomly colored (distractor) words were drawn on top. All words are randomly positioned in the image. Finally, the background for each image is chosen randomly from the hue spectrum, but with a reduced brightness of 50%. (**B**) Error proportion as a function of hue. Separate output layers have been trained on a set of 7 color bands that are shifted from the left to the right border in 10 steps for each category (with 15 repetitions per step). This means, that while one network is trained to classify words of colors sampled from a narrow range on the left side of each category, another network is trained to classify words of colors sampled from the right side of each category for each respective class. After training, the performance is evaluated using novel samples on the hue spectrum that match the color bands the network is trained on. Subsequently, the resulting error rate is displayed in the colored line by combining the performance for all the network instances (shaded grey region represents one standard deviation). The black dotted vertical lines indicate the category bounds found in the original Invariant Border Experiment. In this manner, we can see the error rate typically increases as it approaches a border.

important to recognizing elements in our visual world (*Witzel and Gegenfurtner, 2018*). The notion that color categories emerge as a result of acquiring a basic visual task such as object recognition is also in line with the finding of universal tendencies in the development of color categories across cultures (*Kay and Regier, 2003*). Further exploring the origin of the borders, the results shown in *Appendix 5: K-Means Clustering* indicate that efficiently representing the distribution of colors in the ImageNet database we used might, in part, explain the borders, in line with earlier results by *Yendrikhovskij, 2001*.

## Mixed color stimuli

The converging results from the above experiments provide a strong indication that the network represents colors categorically. Still, the stimuli used above deviate considerably from the ImageNet database. We, therefore, wanted to ensure that the estimated borders are directly connected to how the network treats color. We proceed by investigating to what extent the borders generalize to tasks more similar to the one that the network was originally trained on. Here, in a first step, we introduce more complex stimuli that are comprised of multiple, colored, words on a randomly colored background to introduce contrast variations (see *Figure 6A*). Specifically, stimuli were comprised of three words colored based on the training specifications, as well as two additional words that were colored randomly. Finally, the background color is also randomly selected from the hue spectrum, however, at a lower brightness to ensure that there is sufficient contrast between words and background. The task of the network is to classify the stimuli based on the three correspondingly colored words. The colors of the distractor words as well as the background have no predictive value and should be ignored by the network.

The key question is not whether the network can perform this task, but whether the categories obtained above are meaningful in light of more complex color images. Therefore, we trained new classifiers while iteratively shifting training bands from left to right within each category (in 10 steps). This allows us to evaluate performance as a function of the training band positions within the category. Two possible outcomes can be distinguished: On the one hand it is possible that for such complex stimuli the network deviates from the obtained categories and performance does not depend on where we select our training bands in the category. Alternatively, if the system can benefit from the categorical coding of color, we expect performance to be highest (and the error rate to be lowest) at the center of the categories, while the error rate should peak when the training bands align with the borders of the categories. In *Figure 6B*, we see that this latter categorical approach is indeed what the network relies on: Overall, the network is able to perform the task reasonably well, but error rates are lowest toward the category centers, while increasing towards the borders. As in the experiments above, there is a slight divergence at border between green and turquoise.

## Recognizing colored objects

The previous experiment extends our finding to more complex color stimuli. We chose word-stimuli because they are made up of a rich set of patterns with many orientations. One notable element missing in these word stimuli are the large surface areas that are typically found in objects. In this experiment, we investigated whether the previously found categorical borders still guide classification when classifying objects that incorporate large uniform areas. For generating objects with larger uniformly colored areas, we rely on the *Google Doodle Dataset* (*Ha and Eck, 2017*). This dataset includes thousands of examples of hand drawn objects gathered from individual users. Because each drawing in the dataset is stored as a series of vectors it lends itself well to redraw the lines and fill the resulting shape with a uniform color (we plot some examples in *Figure 7A*). To further evaluate the usage of color categories in object recognition of the CNN, we added one additional manipulation. So far, our experiments have aimed at looking on the reliance on color in isolation of potential other factors. However, with the introduction of objects of different shapes, a natural question is to what extent the network uses color or shape(s) to classify the objects? To obtain a better insight into the interaction between these components, we also raised the number of classes from 7 to 14. This allows us to evaluate whether the network simply ignores the color categories when they are not the sole source of discrimination, or can use them in combination with shape features.

We ran 100 iterations, for each selecting a random permutation of 14 objects and assigning them to the 14 constant training bands on the HSV hue spectrum (note that this results in 2 object classes

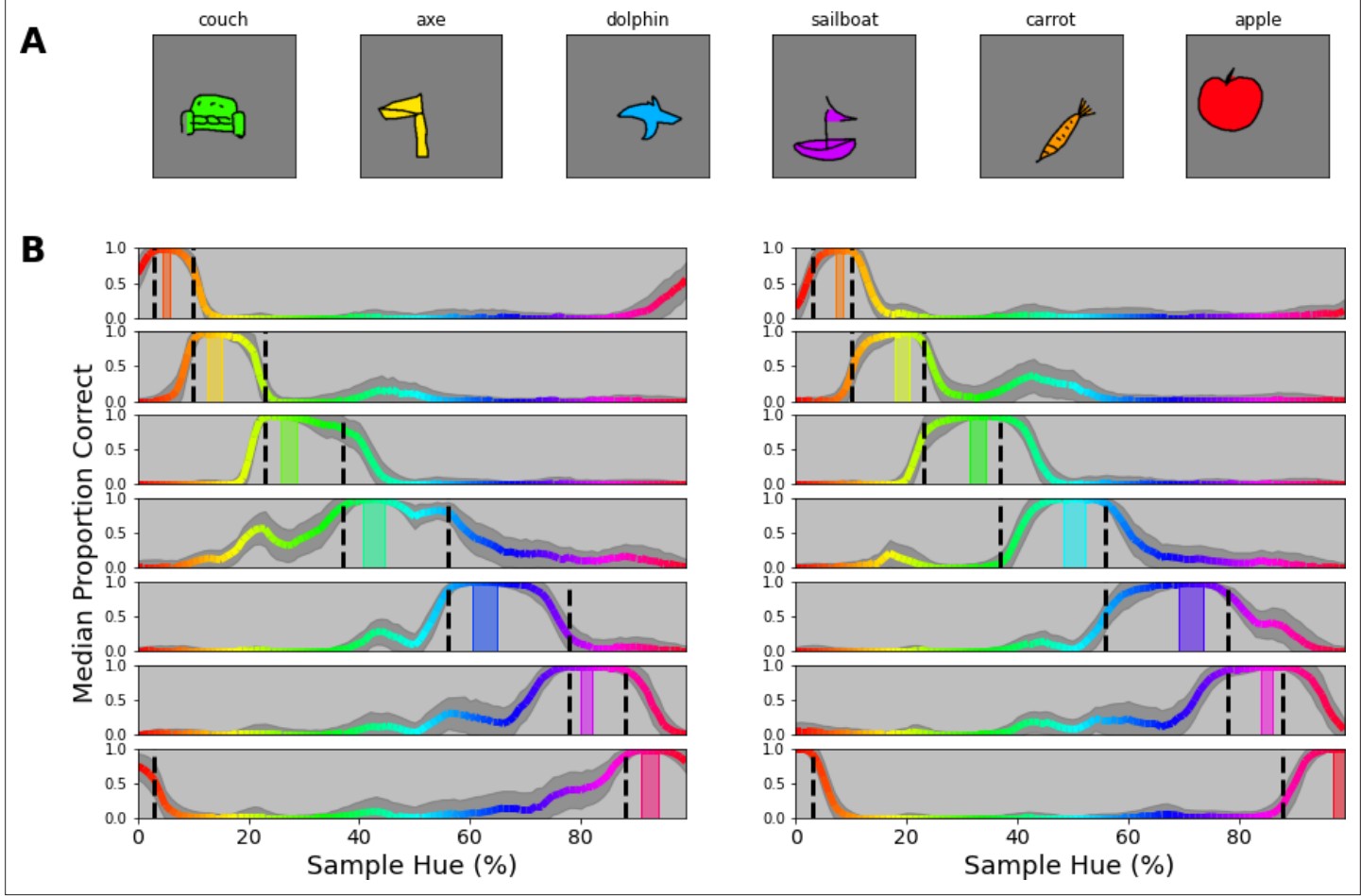

**Figure 7.** Colored objects experiment. (**A**) Samples of the google doodle dataset as colored by our simple coloring algorithm. (**B**) Proportion correct as a function of hue. The 14 individual plots correspond to the 14 training bands that have been selected, 2 per category, one to the left of the category center, another to the right. The training bands are indicated by transparent rectangles on the spectrum, colored for the center of the band. The network's output layer is retrained 100 times, each time with a different permutation set of the 14 objects. For each permutation, we evaluate performance for each object, matching the training band. Per instance, 80 drawings of the respective object are filled with each color of the hue spectrum (over 100 steps) and proportion correct is averaged for each object over the hues. The colored lines represent the median performance on this evaluation (over the 100 iterations), with the hue of the line representing the color in which the object was evaluated. The shaded grey area indicates the standard deviation over the 100 repetitions.

per category). After training, the newly trained output layer is evaluated by taking examples of each of the 14 objects and filling them with colors from over the entire hue spectrum (again divided in 100 steps). In *Figure 7B* we plot the results for each of the 14 bands, separately. We observe that performance is consistently high within the category borders in which the training band falls, while in most cases there is a steep drop in performance outside of the category bounds. This means that on the one hand the network uses color to distinguish objects from those that are colored for different categories. At the same time, however, it appears that to discern two objects within a category, classification relies on the object shape. As such, the network appears to combine both color and shape information and, important to the current research question, the representation of color it relies on, closely follows the previously found category borders.

## Discussion

The relation between cognitive concepts and language has sparked a lively debate (e.g. *Casasanto, 2008*). In this light, the potential relation between language and color categorization has been extensively researched as a prime example of categorical perception. A number of recent studies on color

categorization have focused on the development of categories by explicitly modeling their shape utilizing communicative concepts. Here we have taken a different approach by evaluating whether color categorization could be a side effect of learning object recognition. Indeed, we find that color categorization is an emergent property of a Convolutional Neural Network (CNN) trained for object recognition. Notably, we show that the final layer of a ResNet-18, where objects are represented for classification, reveals a color representation where colors are organized in categories similar to those of humans. In contrast, the representation of color in a ResNet-18 trained to classify scenes (distinguishing man-made from natural) does not follow the hue spectrum and deviates considerably from the human categorical organization. The fact that task modulates the color representation in the final layer of the CNN, where objects are represented, suggests an important connection between object recognition and color categorization.

Importantly, our findings are in line with most of the previous research on color categorization. First, the notion that color categories arise based on an interplay between the distribution of colors in the visual world and basic visual tasks is in line with the notion that the emergence of color categories over cultures broadly follows a universal pattern. However, given that the current study does not directly test this assumption we should emphasize that we can only conclude that the current results do not conflict with the notion that categories emerge similarly over cultures during the forming of the visual system. Second, the findings are in line with the notion that color categorization emerges in pre-linguistic infants and animals. Third, the findings are in line with recent reports showing a dissociation between color naming and color categorization. Fourth, while many of recent studies highlight how communicative concepts can be used to model category shapes, this correlates strongly with the utility of colors regarding objects. As such, the predictive power of these models may not rely on communication about colors per se, but on taking into account the varying utility over colors. Therefore, it is likely that the implicit incorporation of color patterns useful during object recognition leads to a categorical representation of color in the object representation of the CNN.

The discussion on the genesis of color categorization has a long history, with, for instance, Gladstone already noting the deviating use of color terms in the Homeric poems where the sky is referred to by many analogies (e.g. copper, iron, starry), but never blue (see section 4 of *Gladstone, 1858*). Or the suggestion that color categorization is culturally dependent by *Ray, 1952*. The current scientific debate draws strongly on the seminal work by *Berlin and Kay, 1969* who proposed that color categories across a wide range of languages could be described using 11 basic categories and advocated that one universal process guides the development of categories across cultures. While this hypothesis was initially based solely on visual inspection of collected categorization data, later, evidence was provided by demonstrating statistical regularities across cultures (*Kay and Regier, 2003*; *Regier et al., 2005*). The current findings are in line with the notion that the general development of categories is similar across languages: When we consider color categorization to be a side effect of acquiring basic visual skills (given relatively similar circumstances across the globe) color categories are expected to shape in a similar fashion throughout many cultures. Naturally, the current findings do not preclude the notion that categories develop differently in cultures. If categories emerge based on acquiring basic visual skills, we should also see local differences based on different color distribution in the objects important in cultures. An important next step to verify that there is any explanatory value to the current results, would be to gather datasets specific to different cultures and evaluate whether training networks on similar tasks leads to categories that align with the local culture.

Measuring the full hue circle in pre-linguistic infants, *Skelton et al., 2017* found that their novelty preference was dependent on categorical crossings. As such it appears that some form of categorical representation develops early on. However, it is unclear whether these categories are the direct basis for categories in verbal communication. Four of the five categorical distinctions Skelton and colleagues found can be separated by the cardinal axis corresponding to the color representation in the retinogeniculate pathways, suggesting some of the categorical behavior may rely on the early representation of color (an issue pointed out by, e.g. *Lindsey et al., 2010*; *Witzel and Gegenfurtner, 2013*). Also, color naming develops much later and children that have not yet acquired color names make recognition errors based on perceptual difference (*Roberson et al., 2004*). Similarly, it is unclear whether categorical representations of color in animals resemble those in humans. Nevertheless, recent findings have shown, that, as in humans, the utility of color may play an important role for the categorical color representation in animals (*Caves et al., 2018*; *Hanley et al., 2017*). As such, while

the nature of the categorical representations in pre-linguistic infants and animals is still somewhat unclear, the novel finding that a categorical perception emerges with the general acquisition of visual skills is in line with the recent findings. Moreover, it is likely that any pre-linguistic categorical behavior lays the basis for subsequent color naming.

As in animal studies, we relied on a match-to-sample task for studying color categories in a CNN. With this indirect approach, some limitations are similar to those in studies on animals. Importantly, however, there are also clear advantages to studying categorization in a CNN. We were able to repeat the match-to-sample task for a great number of training colors (without the risk of inducing a sequential bias over training sessions) allowing for better estimates. Moreover, verifying the border locations using the concept of categorical perception with an evolutionary algorithm is a computationally intensive task that cannot be straightforwardly applied to any living system, but is feasible for CNNs because of the speed at which an output layer can be retrained. Of course, compared to animal and infant models the CNN is the least convincing version of the adult human visual system. Nevertheless, an important benefit is that the access to activity of artificial neurons is complete and can be exploited in future research: While neural networks are often described as black boxes, compared to biological systems, artificial neural activity is much more accessible, as one can easily probe the activation of all neurons. With this kind of access, a logical next step is to investigate how the obtained categories are coded in the CNN. Coding of colors seems to become narrower and more variable beyond the LGN (as demonstrated in macaque studies by *Conway et al., 2007*; *Kiper et al., 1997*; *Lennie et al., 1990*) and colors belonging to the same category seem to be clustered together in human visual cortex (*Brouwer and Heeger, 2013*). *Zaidi and Conway, 2019* suggested such narrowing may take place over areas (from V1 to IT) by combining earlier inputs (equivalent to a logical AND) or through clustering of local cells. CNNs can serve as a model for testing the viability of several of such concepts.

Where the terms color categorization and color naming are often used synonymously, the subtle distinction between them is key to the debate on the emergence of color categorization. From a strong universalist point of view, a color name is no more than a label applied to perceptually formed categories. From the relativist point of view, the direction is reversed and it is the existence of a color term that dictates the existence of the respective perceptual category (*Jraissati, 2014*). A recent case study shows a dissociation between color naming and categorization. Patient RDS is able to categorize colors, but his color naming ability is impaired (*Siuda-Krzywicka et al., 2019*; *Siuda-Krzywicka et al., 2020*). The dissociation between the two, favors a view where categorization is a process that can exist independently of linguistical labels. Interestingly, despite the problems in color naming, the link between objects and colors was preserved in RDS (*Siuda-Krzywicka et al., 2019*). While it is possible to argue that our CNN does communicate (as it assigns objects to a class), it is important to note that the CNN at no point is required to communicate about colors directly, but at best about objects. However, there is no exchange (back and forth) of information between multiple CNNs, rather a single CNN merely imparts information. Therefore, while the current findings could be viewed in terms of communication indirectly shaping color categories, they only show a very limited overlap with the hypotheses linking category formation and communication in current literature. As such, we argue that the most important part of the current finding is that it demonstrates a direct connection between the formation of color categories and learning to identify objects.

The fact that the current categorical representation appears to emerge in the absence of color naming emphasizes that explicit color naming is not a necessity for the development of categories. This may seem to stand in contrast to many of the recent studies that use communicative concepts as a means to model the shape of categories (*Chaabouni et al., 2021*; *Twomey et al., 2021*; *Zaslavsky et al., 2020*). However, many of those studies derive their predictive power from combining these concepts with the non-uniformities in the utility across colors. As such, the communicative concepts could merely be a means to incorporate the varying utility across colors. Previous studies have demonstrated that the utility of colors as they relate to objects is reflected in the notion that warmer colors are communicated more efficiently than cooler colors (*Gibson et al., 2017*), it has also been shown that objects are associated with warmer colors than backgrounds and that classifiers trained using only color information can distinguish animate from inanimate object at a performance level comparable to using shape features (*Rosenthal et al., 2018*). The latter emphasizes that the higher communicative need for warmer colors, likely stems from their prevalence in objects. While we do not argue the process is completely devoid from communicative factors, the current results can unify many previous

findings by showing that acquiring a skill like object recognition can lead to the emergence of a categorical representation of color.

## Methods

### Invariant border Experiment

#### Software architecture and stimuli

The experiment utilizes a ResNet-18 as provided in the *models* module from the *torchvision* package (**Marcel and Rodriguez, 2010**). The network is initialized with the *pretrained* option on: weights are set based on having been trained on ImageNet (**Deng et al., 2009**) a large database of human-labeled natural images. After initializing the network, we replace the output layer (that performs the object classification) with a smaller output layer with anywhere from 4 to 9 output nodes. The weights to this novel (replacement) classification layer are randomly initialized.

Stimulus images are generated using the *Pillow* package (**Clark, 2015**). Image size is the same as that used for the original ImageNet training (224x224 pixels). Each image contains the word 'color' randomly positioned on a mid-grey background (font size 40, approximately 100 by 25 pixels, depending on font type). The color of the word is randomly (uniform) selected from the bands on the hue spectrum in HSV space. Color bands representing the individual classes are uniformly distributed over the hue space (see *Figure 1A* for stimuli sampled from example bands in *Figure 1B*). Brightness and saturation are set to the maximum level. The HSV color for each pixel is converted to its equivalent in RGB space as the network has been trained using three input channels for red, green and blue, respectively. The font of the word is selected from 5 different fonts. Stimuli are normalized using the same parameters across all images (the same values as used for the original normalization of ImageNet images). Note that this normalization is purely done to have a range with 0 mid-point, making the numerical computation more stable and should not affect the colors as the channels are independently fed to the network and multiplied by learned parameters.

*For the scene-task*, where natural scenes are distinguished from artificial scenes, we used the classification in the subclass of ImageNet that distinguishes natural from artificial scenes. Otherwise, the analysis was the same as for the object-trained network.

#### Procedure

For each number of output nodes (4 through 9), we instantiate the network 150 times, replace their output layers and train each on a slightly shifted set of training bands. The combined width of the training bands equals 20% of the total hue range. During network training we only allow the weights of the new classification layer to be updated, the weights of all preceding layers remain as they were trained on ImageNet. Because we cannot determine the number of potential color categories a-priori, we vary the number of output classes from 4 through 9. This results in training a newly initialized output layer for 150 (band shifts) times 6 (4 through 9 output classes) networks, making for a total of 900 training sessions. 500 samples are provided for each class and the network is trained for 5 epochs. During the training, we keep track of the best network using 50 separate validation samples per class (from the same training bands). After training the network to classify the colors from the training bands, each network is evaluated over the whole hue spectrum by providing the network with 60 samples for each step on the HSV hue spectrum (divided into 100 steps). This results in 6000 classified samples for each of the 900 trained networks.

#### Analysis

The 6000 test samples for each trained output layer are used to determine the border crossings. In *Figure 1C* we plot the classification of these 6000 samples for a single training iteration. To determine the best prediction, for each step on the hue spectrum (each column in *Figure 1C*), we take the mode. In this manner, we transition to a one-dimensional representation of the network's performance on the evaluation task, with the prediction for each hue. Importantly, this one-dimensional representation, as plotted in *Figure 1D*, is used to determine the border crossings: for each network we determine the borders by simply picking the transition between predicted classes. Finally, we sum all the borders and from this we use a simple straightforward peak detection algorithm (findpeaks from the scipy.signal library) to find the locations on the spectrum where the borders are most invariant to change.

To determine how 'categorical' the found border invariances are, we determine the maximum cross-correlation for each row compared to every other row, by shifting one of the rows and finding the optimal shift by looking for the maximum cross-correlation. To ensure the circular nature of the hue is preserved, hues are converted to 2D locations on a unit circle and a 2D cross-correlation is run for the x and y coordinates. By obtaining the shift for each row compared to all other rows, we obtain a distribution of shifts, that can be compared to distributions representing a categorical result and continuous color result. The former is generated by training a ResNet-18 (from scratch, i.e. initialized with random weights and allowing for the parameters of all layers to be updated) on the currently obtained categories and, subsequently, evaluating it in the same manner as we evaluated the ResNet-18 trained on ImageNet. The latter distribution is determined by calculating the optimal shifts for the case where the borders between colors move in parallel to the shifting bands.

## Human psychophysics

### Observers

Ten observers were included in the experiment (of which 5 male and 5 female) with an average age of 32.1 (SD: 4.3). Two of the observers are also authors (JV and AA). Informed consent was obtained from all observers prior to the experiment. All procedures were approved by the local ethics committee at Giessen University (LEK 2021–0033).

### Stimulus and apparatus

Stimuli consist of one central colored ring accompanied by 7 colored, peripheral, disks placed on an imaginary half circle around the central circle (see *Figure 4A* for an example stimulus). The central ring color is selected from the hue spectrum of the HSV color space divided in 35 steps. The peripheral ring colors are also selected from the HSV hue spectrum in a manner consistent with the original Invariant Border Experiment, meaning that 7 equidistant hues are selected for each trial and they are shifted in 35 steps around the hue spectrum. The experiment is performed on an iPad (9 of 10 observers performed the experiment on the same ipad). A strict distance (eyes-to-center-screen) is not controlled, however, at a typical operating distance of ~50 cm, the eccentricity of the peripheral disks is 5.6 degrees and their diameter approximates 1.8 degrees. The center disk is 2.8 degrees in diameter at the typical operating distance.

### Procedure

Starting the experiment each observer gets to select their preferred iPad orientation (landscape or portrait) to perform the experiment in. The observer is bound to performing the whole experiment in this orientation (changing the orientation of the iPad results in the experiment being paused with a warning). To ensure the observer has ample time to observe all options and avoid choices being influenced by transient or salient effects, the 7 colored peripheral choices are presented throughout the entire trial. With the 7 colored peripheral choices, the observer is presented with a central white circle. Upon placing the finger on this circle, it shrinks to disappearance (over 150ms) after which it is replaced with the ring in the target color. Upon the color change, the observer is tasked to select the peripheral disk that is most similar in color. The observer selects a peripheral disk simply by tapping it with her/his finger. Once the selection has taken place the trial is completed and the peripheral disks change color while being shuffled into novel locations in preparation of the next trial and the central circle reverts to white. For each trial, the disks (with respect of the midpoint) are rotated either slightly clockwise or counterclockwise (15 degrees) to reduce the overlap of the disks between trials. Observers are told to perform the trials in a speedy manner, but aim to maintain a high accuracy (in accordance with their own perception, as there are no predetermined correct responses) over the trials. In total, each observer completes 1225 trials (35 *hue target variations* **x** 35 *shifts in the peripheral disk hues*). To ensure finger fatigue does not negatively affect the results, observers are free to pause the experiment at any point and return to it later.

### Statistical analysis

To ensure the transition count over observers, as presented in *Figure 4C*, is reliable we constructed a statistical test utilizing bootstrapping. Specifically, we created a *null-distribution* of uncorrelated

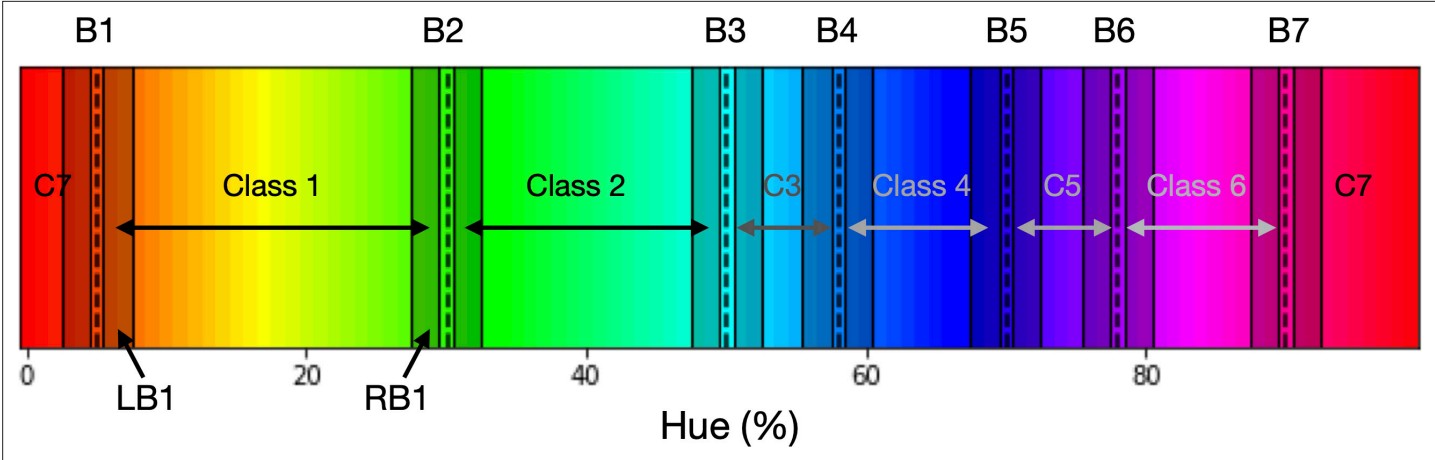

**Figure 8.** A single set of 7 borders (indicated by vertical dashed lines; labeled B1 through B7). Each space in between two adjacent borders represents a class. Colors for the training samples for, for example, Class 1 are randomly selected from one of the two bands, LB1 (Left Band for Class 1) and RB1 (Right Band for Class 1), on the inside of the borders of the class. Each of the two bands (not drawn to scale) comprises 10% of the space between neighboring borders; for class C1 this is the distance between first dotted vertical line (B1) and the second dotted vertical line (B2). Note that this means that the bands for class C3 for example are thinner than for Class 1. To ensure the training bands do not overlap at the borders, there is a gap (comprising 5% of the category space) between the border and the start of the band.

observers by generating bootstrapped datasets where the observers' transition counts were randomly shifted. This null-distribution is then used to compare with the observed correlation in our original observers.

To estimate the number of observers required for this analysis, we used data obtained from the different architectures analyzed (see Appendix 8). From this data, we created two bootstrapped distributions: One null-distribution where the transition counts in the bootstrapped datasets are randomly shifted (as above) and one without random shifts, to obtain a distribution for correlated observers. This demonstrated that with 6 network 'observers', the probability of obtaining a non-significant was already less than 5%. However, to air on the side of caution and allow for potential follow-up analyses we decided to recruit 10 observers.

## Evolutionary experiment

For this experiment we initialize 100 sets of 7 borders, each ordered from left to right (each representing a point-position on the hue spectrum). For each border-set the ResNet-18 pre-trained on ImageNet is initiated and again the final fully connected layer is replaced. The stimuli for training the networks are generated via the same procedure as described in the Invariant Border Experiment, but now the hues for each class are selected from two narrow bands just inside of each set of neighboring borders (see *Figure 8*). Both band positions, as well as width, are relative to the two adjacent borders; The band starts slightly inside the border (with the closest edge located 5% from the border) and the band width is set to 10% of the total distance between the borders. We use two narrow bands at the ends of the potential category as this will mean that when the borders cross a categorical boundary the network will have to learn to generalize colors from different categories to single classes, while if the borders are located optimally, the 2 bands for each class will stem from 1 category each. We judge the fitness of a border set by evaluating how *fast* the network can learn the classes defined by the bands on the inside of two adjacent borders. Therefore, each network is trained for 3 epochs only, which is insufficient to reach peak performance, but allows us to evaluate which border set best fits the color representation of our ResNet-18: Border sets that align with the color representation of the network should allow the network to reach a higher performance quicker.

After training the 100 networks, the border sets are sorted by the performance of their respective network. Note that while we base the ordering on the performance of the network, the network is assumed a constant factor and the performance is attributed to the border set that is used to generate the training set; Essentially the network (trained on only 3 epochs) is the fitness function for the border sets. Once the fitness of each border representation has been established, the sets can be ordered by

fitness and we generate a new generation of border sets. Firstly, through the principle of elitism, the top 10 performers of the current generation are copied directly into the next generation. Additionally, a novel 90 border sets that are created by recombining border sets from the current generation. Specifically, to create a new set, we select two parent border sets and, first, combine any borders that occupy similar positions across the two sets: This is done by averaging borders that are less than 5% apart between the sets (we start by combining the closest borders and the threshold is lowered when many borders are within a 5% range). Secondly, from the resulting borders, 7 are randomly selected to create a new border set. In order to converge to the optimal border-set, the selection of the two parent sets is biased towards the better performing border instantiations in the current generation as follows: 55% of the parents are selected from the 25 best performing border sets; 30% from the next 25 sets; and 15% of the borders are selected from the 25 sets thereafter. The bottom 25 border sets (in terms of performance) do not participate in the creation of offspring. To ensure some exploration occurs, in the offspring, some borders are randomly shifted. Specifically, we randomly select 2.5% of all borders and randomly shift them (random shift is normally distributed with an SD of 2.5% of the hue spectrum). The whole process is repeated 40 times. To allow for convergence after 30 generations random mutation is switched off.

## Multi-color Experiment

### Software architecture and stimulus

Again, the same ResNet-18 trained on ImageNet is used as in the previous experiments. In the current version, the output layer is replaced by one with 7 output classes, each matching a category. The stimuli were designed to include 2 factors that were absent previously. Firstly, we introduce multiple, colored elements in a single stimulus: Each stimulus class is still defined by a narrow color band on the hue spectrum, but we now draw 3 words into a 224-by-224 pixel image (we will refer to these as the *target* words). On top of that we add two additional words to the image of which the color is randomly selected from the hue spectrum (we will refer to these as the *distractor*s). All words are colored using the HSV hue spectrum at maximum brightness and saturation. Secondly, we introduce variations in color contrast. For the categorical borders to be meaningful in classifying stimuli, their utility should not be overly dependent on color contrast (whether a banana is sitting on a green or a brown table, the network should still be able to use yellow to identify the banana). Therefore, we introduce a variation in color contrast by randomly selecting the color of the background from the hue spectrum. To prevent words from blending into the background, we set the brightness of the background to 50%. Stimulus examples can be found in *Figure 6A*.

### Procedure

Each network is trained on 7 classes divided in 7 color bands, based on the borders found in the Invariant Border Experiment. The bands, making up 10% of each category range, are shifted from the left category border to the right border in 10 steps, for each step a novel classifier is trained 15 times to obtain a reliable average. After being trained we evaluate performance for each network: The error rate for the color bands which the network was trained on is obtained by evaluating it for the ticks on the hue spectrum that fall in that range. Subsequently, we plot the error rates for the all networks as a single-colored line (see *Figure 6B*), to demonstrate how performance varies, depending on training bands.

## Objects Experiment

### Software architecture and stimulus

Stimuli are generated using the line drawings available from the google doodle dataset (*Ha and Eck, 2017*). The database contains hundreds of objects; we selected a subset based on two criteria. First, we selected objects that lend themselves to a simple color filling algorithm. This mainly consisted of finding objects that had clear outer borders with large spaces on the inside. Secondly, we prioritized objects that were reasonably consistent with regard to shape. The drawings are created by many different users, and, therefore, the approach could vary significantly. For instance, a user could have chosen to just draw only a cat's head prominently or include its entire body. Of course, such variations are not unlike the variation encountered in the images the network has originally been trained on. However, we are only retraining the classification layer of the network. The latter means that the

network has to rely on previously trained kernels to classify the shape, and a high degree of variation may not be easy to represent when only adapting the input weights to the last layer.

The drawing of the stimuli follows a simple procedure. Based on the drawn lines we create a color mask. This mask is created by determining, for each pixel in the image, whether it is 'enclosed' by drawn lines. Enclosed, here, is defined by having a drawn line to its left (not necessarily directly adjacent), a drawn line above it, to its right and below it. After the colored area is filled, we draw, on top of it, the drawn lines at a thickness of 4 pixels. Example results of the process can be found in *Figure 7A*.

## Procedure

The network is trained to classify a set of 14 objects (strawberry, apple, crab, dog, school bus, cow, dolphin, mushroom, bird, submarine, angel, sweater, sailboat, duck). For the training of each network, we selected 500 samples for each object from the google doodle dataset. We also selected another 50 samples per object to obtain a validation set to monitor the performance of the network throughout training. The fill color for all of these objects is randomly (uniformly) selected from narrow bands on the hue spectrum. Bands are selected to be non-overlapping and having 2 bands per category, one positioned right of the category center and the other left (each takes up 1/5 of the category bounds, one in the center of the right half and one in the center of the left half of the category). This results in 14 bands that are not exactly evenly spaced, or uniformly distributed throughout the spectrum. The bands can be observed in *Figure 7B*. To obtain a reliable estimate of performance on this task and to minimize the effects of the specific object selection we run the experiment 100 times, each with a different permutation of the object set with respect to the 14 color bands.

After having trained the new classification layer of the network on the 14 objects assigned to the color bands, 80 different drawings for each object class are used to evaluate the network. Systematically changing the color of each of these 80 drawings over the hue spectrum divided in 100 steps creates 8000 colored samples, that allow us to evaluate to what extent the object is classified based on its color compared to its shape. Repeating this process for the 14 trained objects (assigned to the respective color band on each iteration) allows us to evaluate to what extent color was used to define the object class.

## Acknowledgements

This work was funded by the Deutsche Forschungsgemeinschaft (DFG, German Research Foundation) – project number 222641018 – SFB/TRR 135 TP C2, and AF was partially funded by a VISTA postdoctoral fellowship.

# Additional information

## Funding

| Funder | Grant reference number | Author |
| --- | --- | --- |
| Deutsche Forschungsgemeinschaft | 222641018 SFB TRR 135 | Jelmer P de Vries<br>Arash Akbarinia<br>Alban Flachot<br>Karl R Gegenfurtner |
| York University | VISTA postdoctoral fellowship | Alban Flachot |

The funders had no role in study design, data collection and interpretation, or the decision to submit the work for publication.

## Author contributions

Jelmer P de Vries, Conceptualization, Data curation, Software, Validation, Visualization, Methodology, Writing – original draft, Writing – review and editing, Formal analysis; Arash Akbarinia, Alban Flachot, Software, Validation, Methodology; Karl R Gegenfurtner, Conceptualization, Resources, Funding acquisition, Methodology, Project administration, Writing – review and editing

## Author ORCIDs
Jelmer P de Vries (ID) http://orcid.org/0000-0002-3000-9685
Arash Akbarinia (ID) http://orcid.org/0000-0002-4249-231X
Karl R Gegenfurtner (ID) http://orcid.org/0000-0001-5390-0684

## Ethics
Informed consent was obtained from all observers prior to the experiment. All procedures were approved by the local ethics committee at Giessen University (LEK 2021-0033).

## Decision letter and Author response
Decision letter https://doi.org/10.7554/eLife.76472.sa1
Author response https://doi.org/10.7554/eLife.76472.sa2

---

## Additional files

### Supplementary files
• Transparent reporting form

### Data availability
The main analyses were computational and performed on ResNets from the models module of the torchvision package for python (see https://pytorch.org/vision/). Only Figure 4 is based on human data. Network and human data together with source code for running all analyses and generating figures can be found at: https://github.com/vriesdejelmer/colorCategories/, (copy archived at swh:1:rev:b3aa5823d5cb7d04f85e09248bdefeecb5badf84). The code for the iPad experiment is available at: https://github.com/vriesdejelmer/ColorCoder/, (copy archived at swh:1:rev:bd42b08c1ca57c20a401c6c621a2d58668df9e42).

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

## Appendix 1

### Network repetitions

We have chosen to focus on the ResNet-architecture as it introduced residual connections that allowed for creating networks that were much deeper than before. Furthermore, ResNet is uniquely used as the backbone of several important deep networks, vision-language models (CLIP), semantic segmentation (FCN and DeepLab), and Taskonomoy networks. It is also frequently used to compare with human performance in literature (*Geirhos et al., 2018*; *Huber et al., 2021*).

To ensure that our interpretation of the peaks in the transition counts as invariant borders in the hue spectrum rather than noise is correct, we re-run the Invariant Border Experiment a number of times. To also ensure the result is not dependent on network depth, we use the other ResNets available in the *models*-module from the *torchvision* package. Specifically, we repeat the experiment with a ResNet-34, a ResNet-50 and a ResNet-101, all of them pretrained on ImageNet. Moreover, we include a second ResNet-18 trained on ImageNet to ensure the original result replicates. The results are plotted in *Appendix 1—figure 1* below, *Figure 1A* repeats the results from the ResNet-18 as presented in the main text for comparison. Comparing the transition count from the newly trained ResNet-18 as well as the ResNets of different depths (*Figure 1B-E*), we clearly see a very similar pattern repeat. Most borders occur in approximately the same locations across the networks, only slight variations are seen as in some cases an extra border is found (around green or magenta) or a border is missing (between red and orange).

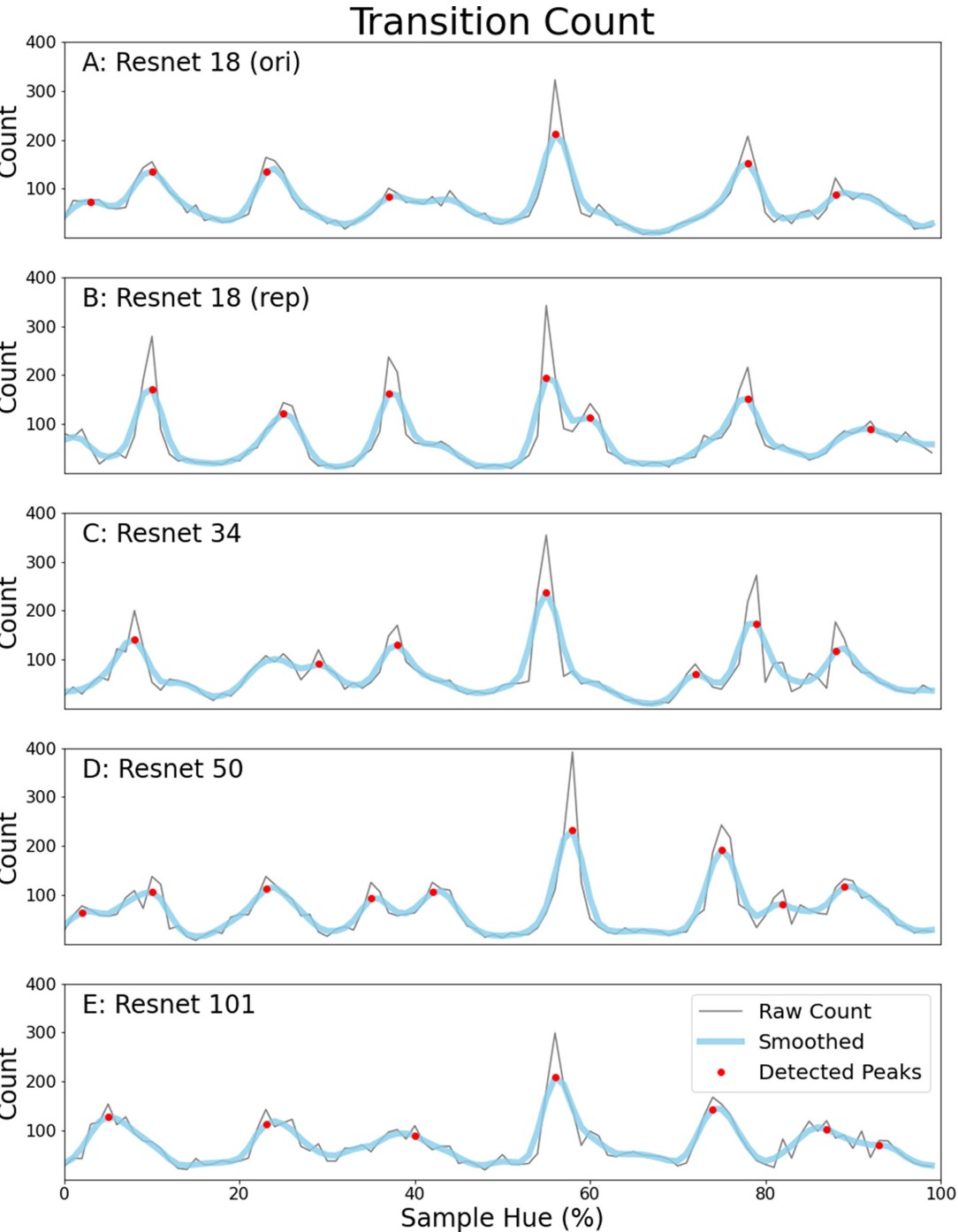

**Appendix 1—figure 1.** Transition counts for five different ResNet instances. (**A**) Transition count for the ResNet-18 from the Border Invariance Experiment in the main text. Transition counts are calculated by summing all transitions in the network's color classifications going around the hue spectrum (see the main document for more details). The thin grey line represents the raw counts, to remove noise we have also included a thicker light blue line which is a smoothed version of the raw count. Detected peaks are indicated using red dots. (**B**) Identical ResNet-18 architecture and dataset, but different training instance. (**C–E**) Same architecture but deeper ResNets all trained on ImageNet: ResNet-34 (**C**), ResNet-50 (**D**) ResNet-101 (**E**).

## Appendix 2

### Simulation classification

To determine to what extent the classifications of the ResNet-18 on the color task can be seen as categorical we compare the classifications to two alternative cases. On the one hand we have the case where borders shift with the training bands, as could be expected when colors are represented in a (quasi) continuous manner. The shifting border simulation is plotted below in *Appendix 2—figure 1* on the left. On the other hand, we have the case where the network relies on a true categorical representation. For this case we have trained a ResNet-18 from scratch on the categorical borders found in the Invariant Border experiment and repeated the same analysis as in the Invariant Border experiment. The result can be seen in *Appendix 2—figure 1* in the right plot. In the middle we have plotted the result from the original Invariant Border experiment for reference. As can be seen, the object-trained and the categorically-trained network produce very similar results, with the exception that the categorically trained result looks less noisy. A quantification of the differences can be found in *Figure 2D* of the article.

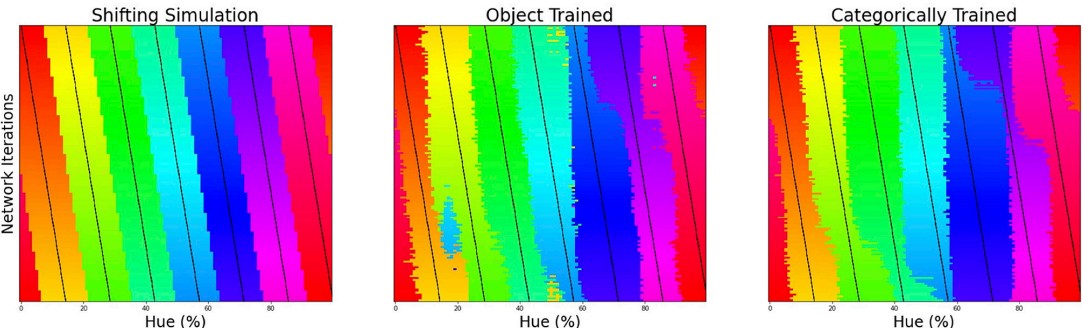

**Appendix 2—figure 1.** Classification simulation. We plot the color classification for the 3 cases from left to right (Shifting borders, data from the Invariant Border experiment and data from a categorically trained network, respectively). Each row in the subplots represents the classifications of a single output layer trained on 7 bands, the center of which is marked by a black tick (appearing as black diagonal lines throughout the image).

# Appendix 3

## Luminance variation

In the invariant border experiment, we have drawn colored words on a uniform grey background. While varying the target word over the hue spectrum of the HSV color space, its luminance also changes. From this perspective it is possible that the obtained borders stem partially from a variation in luminance, rather than color. To ensure this is not the case, we repeated the same experiment as before, but now with two important stimulus adaptations. Firstly, 10 greyscale words (random luminance) have been drawn into the background. Secondly, the background luminance is also randomly selected [range: 80–175 out of 0–255]. Stimulus examples are shown in *Appendix 3— figure 1A*. Otherwise the experiment is the same as the original.

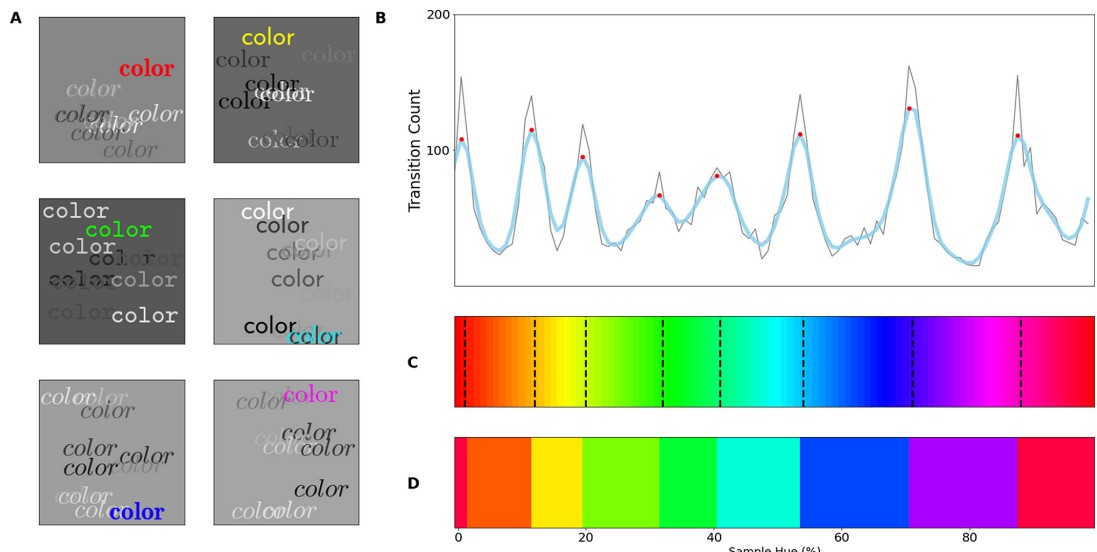

**Appendix 3—figure 1.** Luminance controlled stimuli. (**A**) Stimulus examples drawn from 6 training bands aligning with the primary and secondary colors. (**B**) Raw transition count, that is, the number of times borders between colors are found in a specific location as the network is trained on different training bands is plotted in grey. A smoothed version is plotted in light blue. The found peaks (based on the raw count) are indicated in red. (**C**) Found peaks indicated in the hue spectrum using black vertical dotted lines. (**D**) Average color for each category. The average is weighted by the reciprocal of raw transition count, making the troths in the data the most heavily weighted points.

The results are depicted in *Appendix 3—figure 1B-D*. While the locations of some borders are slightly shifted compared to the original borders, overall, we find a very similar pattern to those of the original experiment. As such, adding luminance variation does not change the notion that a categorical representation of color seems to underlie the current results.

Since the network can rely only on kernels coding purely for color and not a combination of color and luminance to perform the task, it is not surprising that the evaluation produces borders that are slightly different from those when luminance variations are not included.

## Appendix 4

### Circular color spectrum

The hue spectrum from the HSV color space (at maximum brightness and saturation) follows the edge of the RGB color space (see *Appendix 4—figure 1*). To evaluate whether this irregular shape is a factor in the current results we repeat the experiment using a different hue spectrum. Specifically, we rerun the original experiment using colors picked from a single plane in RGB space, all at equal distance from the center (see *Appendix 4—figure 1* for a contrast with the original hue spectrum). The novel hue spectrum is defined as a circle with a maximum radius (staying inside the RGB cube) in the plane of $R+B+G=1.5$. Repeating the original experiment (with a colored word on a uniform background) still shows a certain degree of categorical representation, but, looking at the raw signal count, the result is clearly noisier than the original experiment. Therefore, we have also repeated the experiment using the stimuli from Appendix 3 above. Adding luminance variation, we find a stronger correspondence between the borders from the HSV hue spectrum and the hue spectrum that follows a circular shape in RGB.

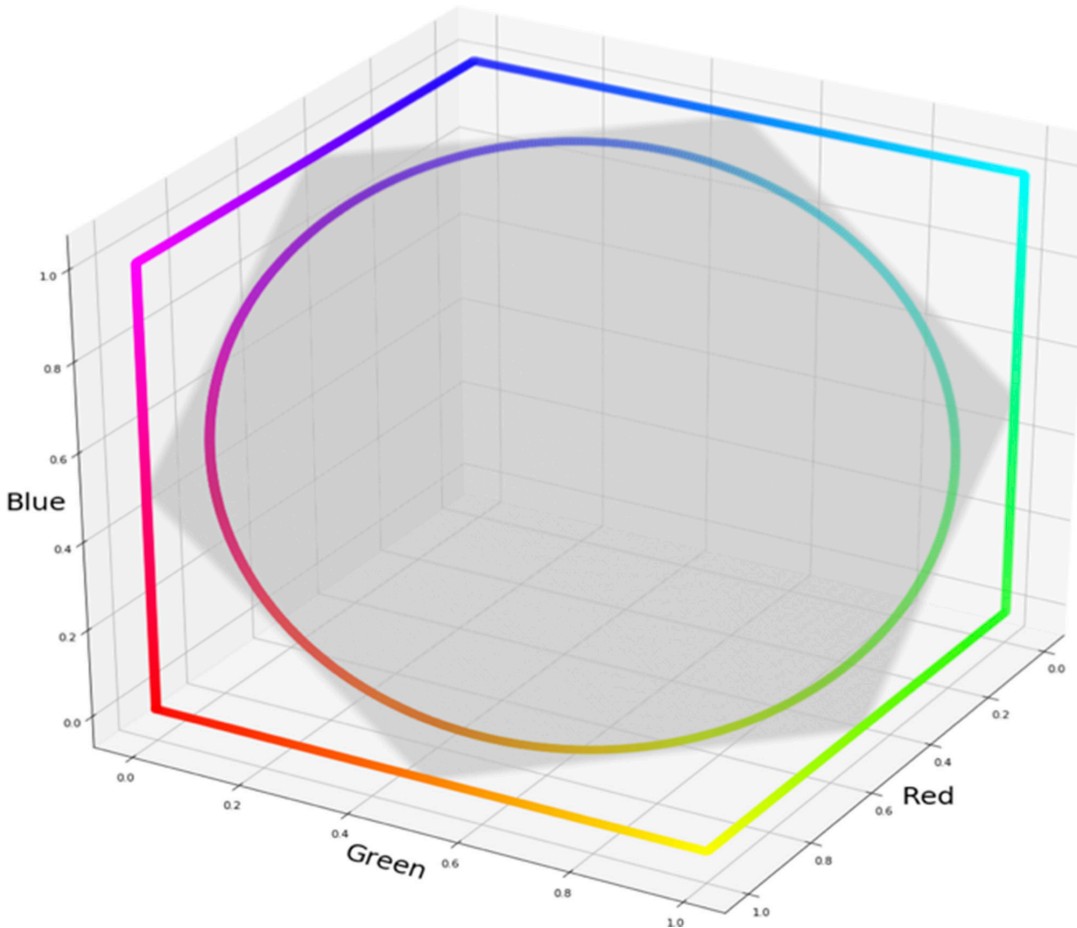

**Appendix 4—figure 1.** HSV hue spectrum and RGB hue spectrum displayed relative to RGB color cube. HSV hue spectrum at maximum brightness and saturation can be seen following the edges of the RGB color cube (subsection of RGB space in which values fall between 0 and 1). The RGB hue spectrum is defined as the maximum circle in the plane $R+G+B=1.5$ (plane is indicated in transparent grey).

Using single word stimuli from a hue spectrum following a circular shape through the RGB cube leads to a slightly noisier categorical representation (*Appendix 4—figure 2A-C*), presumably due to the reduction in chromatic contrast. Repeating the experiment with the stimuli containing luminance variation, however, does result in the same borders as with the HSV hue spectrum (*Appendix 4—figure 2D-F*). We hypothesize that the difference relies on the different kernels the network can rely on to perform the task. Likely, when getting closer to the center of the RGB cube there are even

more kernels that classification can rely on. And, therefore, subtle luminance differences can be exploited and borders become less pronounced.

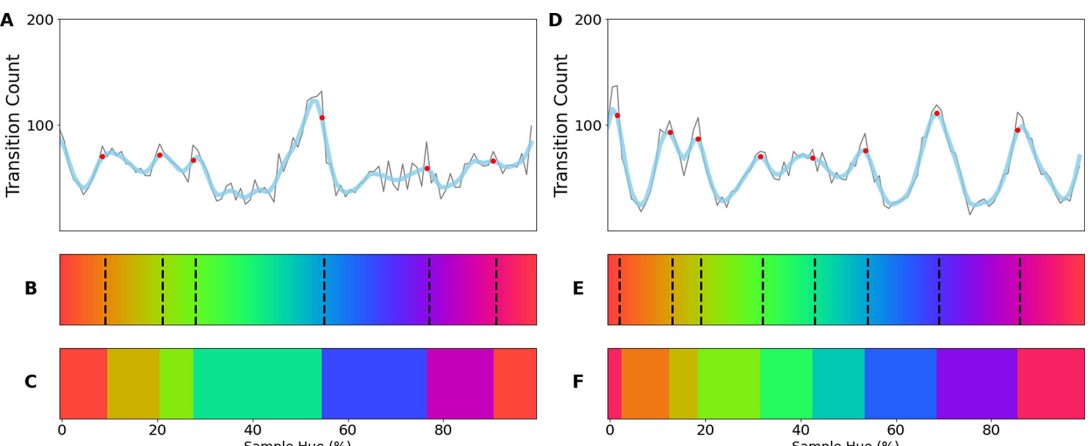

**Appendix 4—figure 2.** Left: Results from rerunning the original experiment with stimuli sampled from the custom RGB spectrum. Right: Results from rerunning the experiment with the stimuli as defined in Appendix 3, but sampling colors from the custom RGB spectrum.

de Vries *et al*. eLife 2022;11:e76472. DOI: https://doi.org/10.7554/eLife.76472

## Appendix 5

### K-means clustering

Given the consistency of the borders over the experiments, the question arises what the origin of the locations of the found borders is. Previously, *Yendrikhovskij, 2001* demonstrated that performing k-means clustering on colors from natural images results in cluster centers that are more similar to the focal color configuration than cluster centers obtained from uniform RGB values. To investigate whether the color distribution of pixels in the ImageNet dataset could explain the location of the current borders, we perform a k-means clustering on them. For this we obtained the frequencies of colors in the HSV hue spectrum from a large sub-selection of images from the ImageNet database (subsection contained images that included bounding boxes; 535,497 images).

K-means clustering is performed by projecting the hues (taken as an angle 0–359) of the pixel values to the unit circle. The 2D coordinates are supplied to the k-means algorithm from the Scikit-learn package (*Pedregosa et al., 2011*) with a count of 7 for the *clusters* parameter. After clustering is performed the cluster means and cluster borders are projected back to the hue spectrum. In *Appendix 5—figure 1* we see that there is correspondence between many of the borders obtained by the k-means algorithm and those found in the convolutional neural network. However, given that the borders do not line up exactly it also cannot completely explain the current border locations. Nevertheless, the correspondence makes it likely that the image statistics drive the location of the borders between the colors.

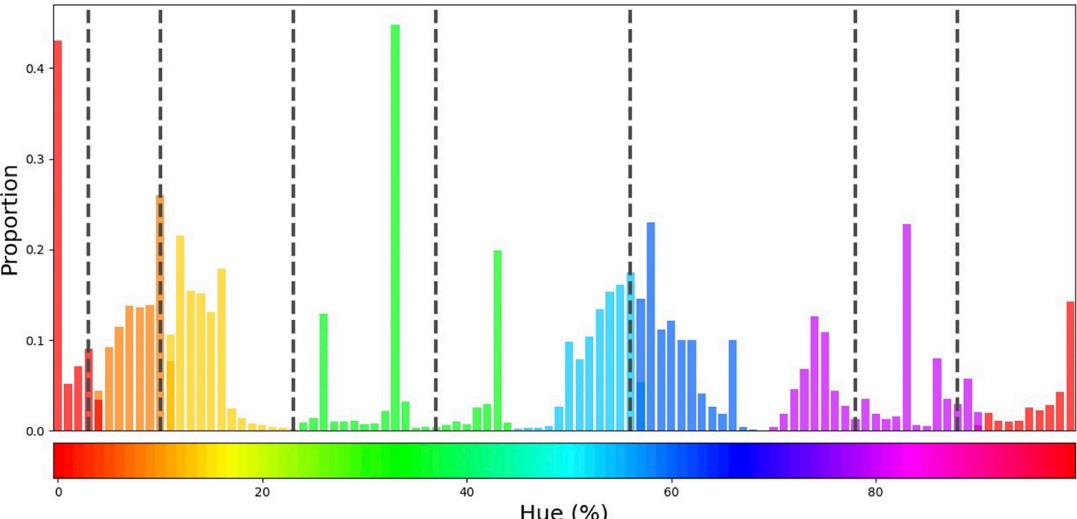

**Appendix 5—figure 1.** Histogram of colors in ImageNet. Colors from ImageNet have been selected to have brightness and saturation exceeding 99%. The bars are colored in seven different colors, corresponding to hues of the centers found by the k-means clustering algorithm. As such, borders between clusters are indicated by color changes in the bars. The black vertical dotted bars indicate the borders as found in the original Invariant Border Experiment.

## Appendix 6

### Layer analysis

It is already known that the structure of color spaces themselves can account in part for the category structures in humans (e.g., *Zaslavsky et al., 2019*). Moreover, the results from the K-means analysis above suggest that the categorical representation in the CNN capitalizes on the variations in the hue distribution in the dataset. At the same time a ResNet trained for scene classification on the same dataset does not incorporate the same categorical representation of color. This emphasizes that even though a color structure can be extracted from the input set, it does not necessarily dictate the color representation in the final layers: Not only does the structure have to be picked up on, it needs to be preserved over many transformations throughout the layers of the network. Given the overlap between the results from the K-means analysis and the categorical representation in the final layer, we expect the categorical representation to be present from the initial layer. To determine whether the representation is already present in the early layers we performed an analysis where we connect a classifier to different components of the CNN pretrained on the object-task: We repeat the original analysis, but now, rather than replacing the final classifier, we attach a classifier to different parts of the network. In this manner, we can see how the color representation evolves through the network.

The ResNet architecture consists of five areas. The initial input area (Area 0) is represented by a convolutional layer of kernel size 7x7, followed by a max pooling layer over a region of 3x3. The subsequent four areas encompass several residual blocks whose skip connections are modeled by summing the input to the output. Within each area, the spatial resolution remains intact. From one area to another the spatial resolution of the signal is reduced by a factor 2 as a result of step size in the convolution operation. We train a new classification layer for each of the different areas of the ResNet-18 in the same manner as we did for the full network to evaluate the representation throughout the layers. In *Appendix 6—figure 1* in the left column we plot the evaluation of the classification when shifting 7 training bands on the right side we plot the transition counts as obtained by summing all transition for the results from 4 through 9 training bands.

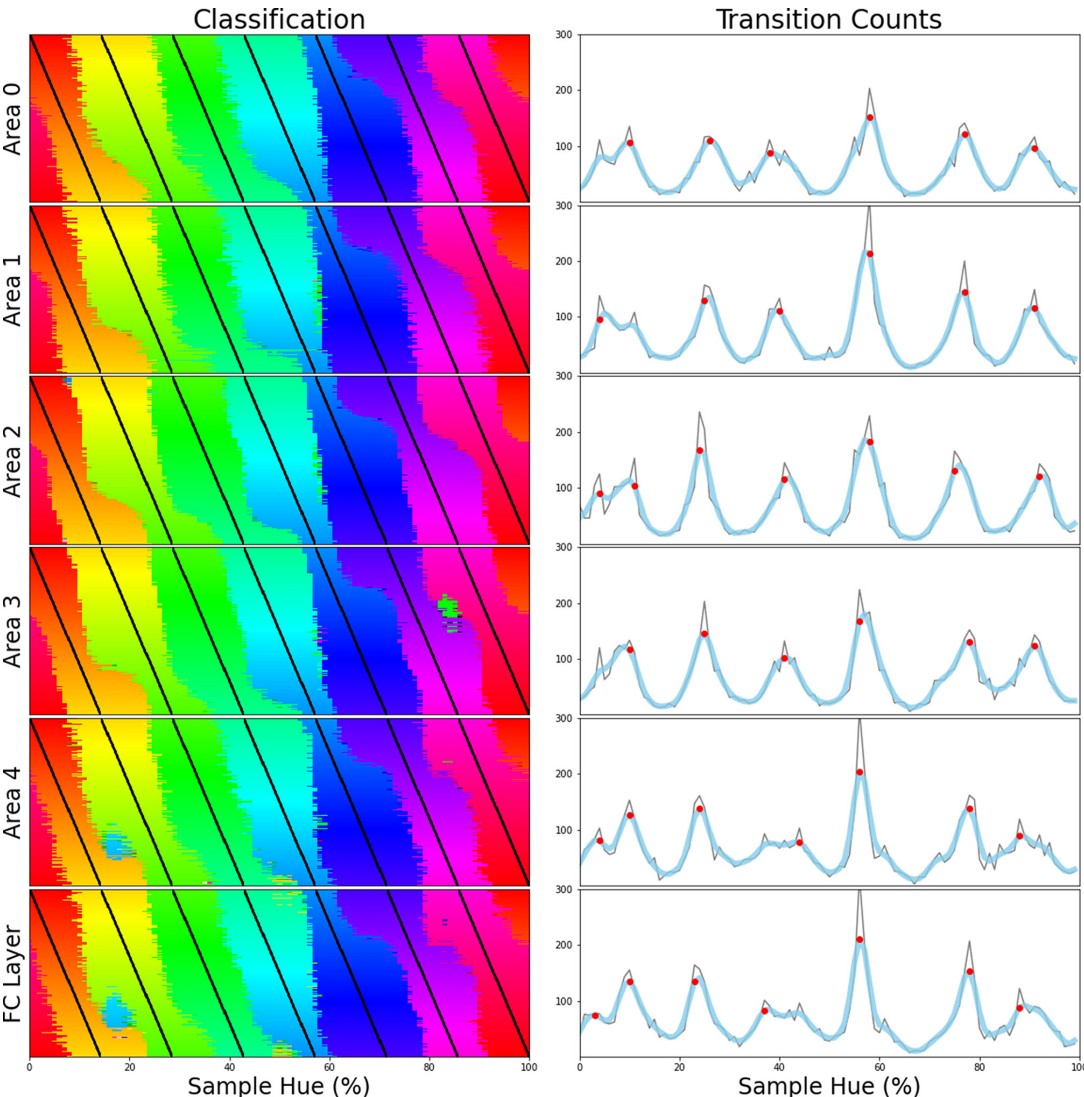

**Appendix 6—figure 1.** Color representations throughout the layers. In the left column each panel shows classification of the network as 7 training bands are shifted through the hue space, as in *Figure 1E* of the main text. In the right column we show the cumulative transition count as accumulated by repeating the process with 4 through 9 training bands. Each row shows the result for a different area of the network with the top row presenting the results for the first area in the network (Area 0). Following rows show the subsequent areas and the final row shows the original result from the fully connected layer of the network. Details can be found in *Figure 2A* of the main text.

The results demonstrate that the representation of color categories is similar throughout the layers and appears to already be present in the initial layer. At the same time, there is clearly some modulation throughout the areas as well; the border between cyan and blue is much stronger in the later areas compared to Area 0.

# Appendix 7

## Random variations

In pursuit of what drives color categorization in the CNN, here, we explore the general tendencies for the representation of color in the ResNet-architecture. For this we analyze two instances with other parameter weights compared to the object-trained version. We train one ResNet-18 while applying a random hue shift to each input, the *random hue network*. In this manner the structure of the input is kept the same (three input channels; RGB), but absolute hue values become meaningless and the network has to rely on contrasts and/or luminance to perform the task. With this network we can evaluate what the effects of training a network on RGB images are, without the ability to rely on hues directly. In a second control we initialize weights of a ResNet-18 randomly, the *random weights network*, and do not train the network on ImageNet. With this network we can evaluate what the general tendency of the CNN is, outside of being trained on object classification with ImageNet.

We display the color representation for the networks with the output layer replaced and trained on 6 shifting training bands in *Appendix 7—figure 1*. Left we show the results of the original experiment for reference. Comparing the performance of the random hue network (middle) we see that as expected color is no longer represented in a continuous manner along the hue spectrum. As such, this continuous representation of color is clearly a result of being trained on a dataset where color is distributed in a meaningful manner. Looking at the classification in the *random weights network* (right) we see that the color representation appears to follow the hue spectrum, but there is a considerable amount of noise, a logical result of propagating activity through the randomly initialized weights. Despite the noise, one could argue that there is still a tendency for straight borders to form.

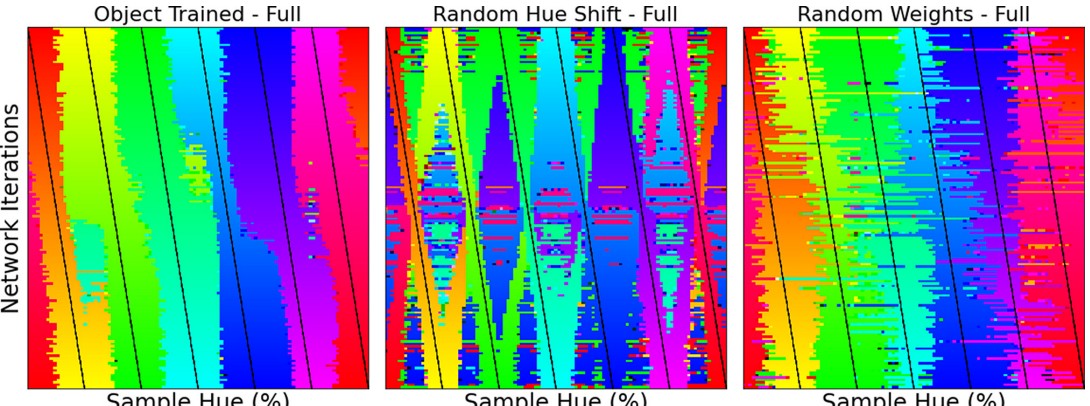

**Appendix 7—figure 1.** Classification of input samples by hue, as extracted from the final layer upon which object classification is performed. Left: The results from the original Invariant Border Experiment for reference. Center: The classification for the random hue network, where the colors in each input image are subjected to random hue shifts. Right: The classification for the random weights network, for which weights have been initialized randomly.

## Appendix 8

### Network controls

The results from the ResNets are consistent. However, in its current state one could argue the paper only demonstrates that color categories are obtained from ResNets, rather than deep convolutional neural networks in general. To see if the result does generalize over CNNs in this section we replicate the experiment with 5 different models, specifically: (1) Alexnet, the first network to score a sub-25% error rate on ImageNet. For efficiency, the network was designed to run two parallel parts on two separate GPUs. Interestingly, it has been found that this parallel design leads to slightly segregated color processing, where the neurons in one stream are more attuned to color, while the other stream includes more neurons attuned to greyscale patterns (*Flachot and Gegenfurtner, 2018*). (2) GoogLeNet, the winner of the 2014 ImageNet competition, based on the Inception network. (3) VGG19 (runner-up in 2014) A large deep CNN with small (3x3) kernels and a large number of channels per convolutional layer. Also notable is that kernels remain small throughout the network. (4) MobileNet V2: a light-weight CNN developed specifically for usage on mobile devices with less processing power. (5) DenseNet, a densely connected CNN, where all layers are connected to all other layers.

The results can be found in *Appendix 8—figure 1*, where the left column displays the transition counts and the right column the classification plots. While the overall results appear similar, several things stand out. The results from the ResNet-18 appear quite similar to those of AlexNet, with the strongest invariance between cyan and blue, and smaller peaks at similar locations distinguishing red, orange and yellow. Overall, the border between cyan and blue appears strongly invariant in all networks. Like ResNet and Alexnet, GoogLeNet appears to have a distinction between red and orange. While all networks, generally, appear to show a categorical structure following the hue spectrum, VGG19 appears an exception to this rule. Potentially this is due to the fact that the network relies on small kernels throughout the network, that may make it more difficult to rely on colors that come in surfaces and may be more focused on simple edges.

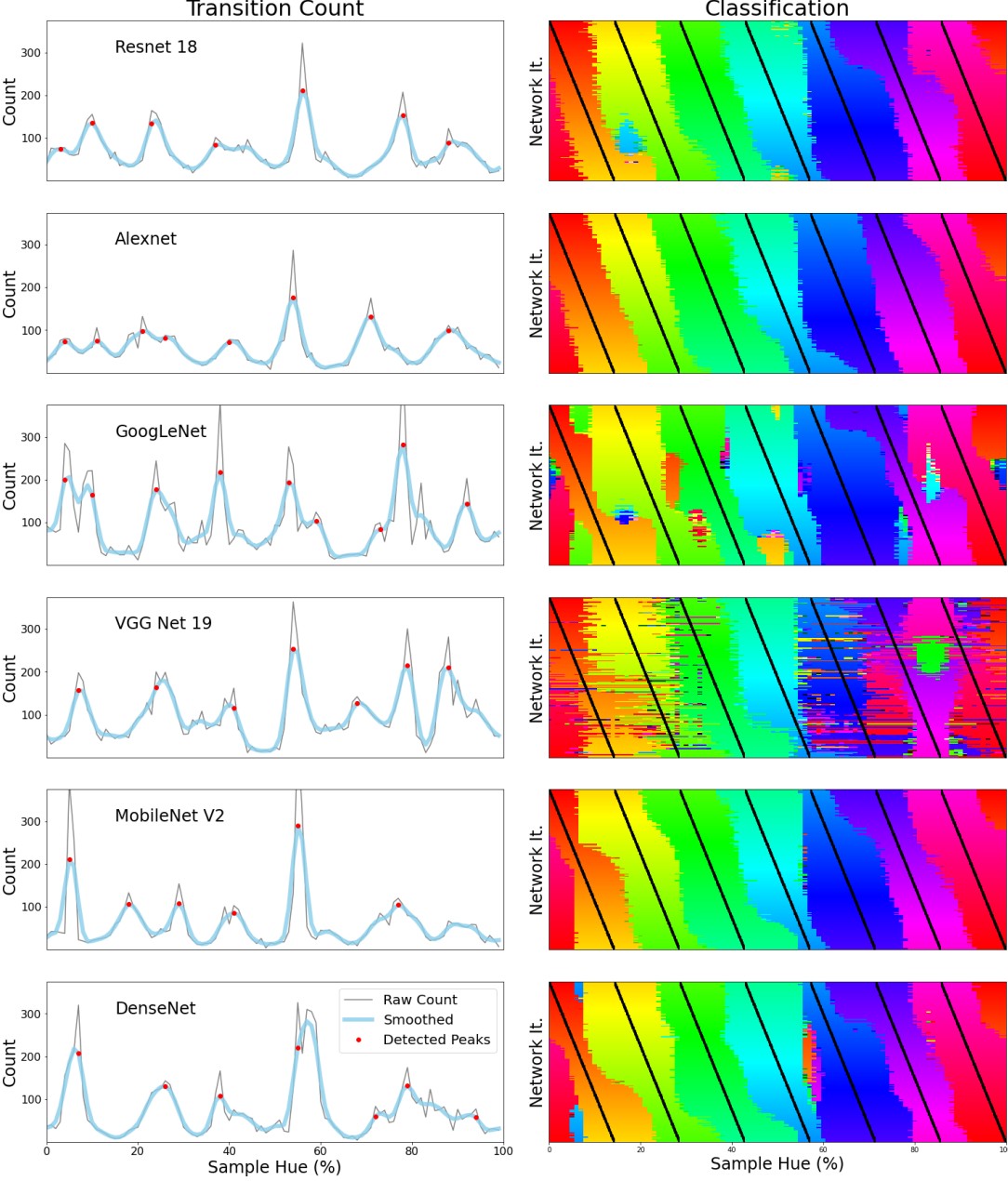

**Appendix 8—figure 1.** Transition counts (left column) and classification visualization (right column) for six different CNNs: From top to bottom: ResNet-18, Alexnet, GoogLeNet, VGG-19, MobileNet V2 and DenseNet. Transition counts are calculated by summing all transitions in the network's color classifications (as obtained from retraining a new output layer for 4 through 9 training bands) and evaluating classification around the hue spectrum. For further details, see the main text; *Figure 2A*. Classification plots are shown for the network trained on 7 training bands. Rows ,in the subplot, show classification for a specific combination of training bands, shifted slightly leftwards for each row. For more details see main text; *Figure 1E*.

## Appendix 9

### Borders along training bands

To provide a complete insight into the color classification and with this the behavior of the borders as different numbers of training bands are shifted through the hue spectrum, in *Appendix 9—figure 1* we include a classification plot for each number of bands used in the experiment in the Invariant Border Experiment. Notably we can see the classification for 4 training bands is noisier, which is likely due to fact that the space between bands is comprised by more than two categories and as such in-between hues are more likely to be generalized to a different class.

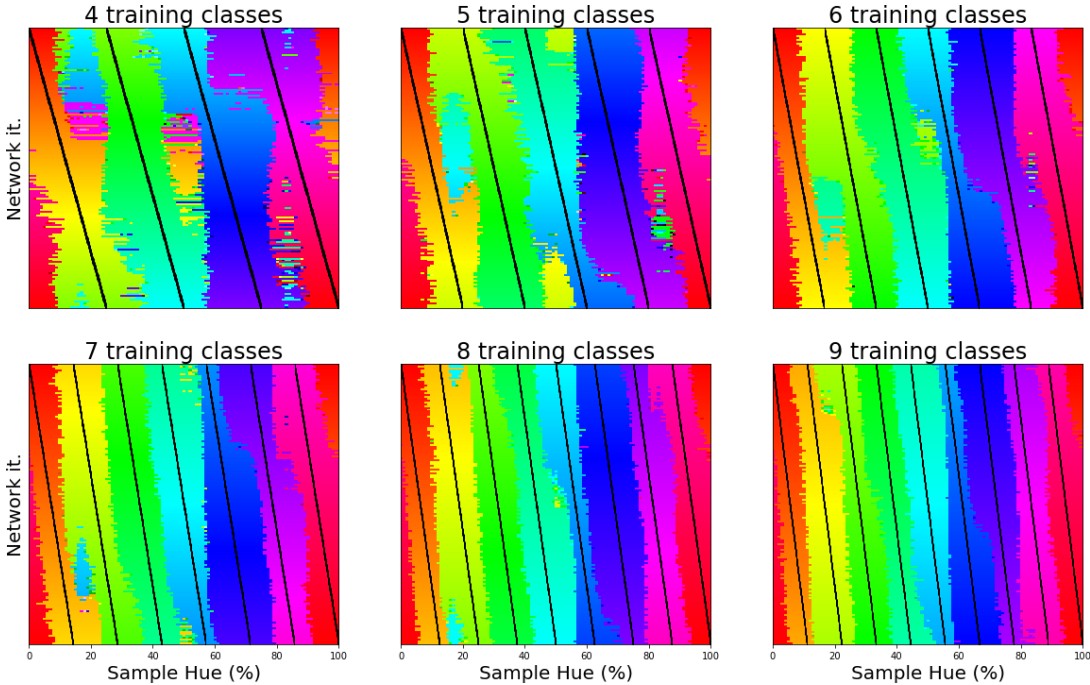

**Appendix 9—figure 1.** Classification plots for 4, 5, 6, 7, 8, and 9 training bands, respectively, ordered left-to-right, top-to-bottom. Specifications of these plots can be found in the main text; *Figure 1E*.

To further explore the notion that borders are invariant to the training bands, we also introduce a post-hoc test for the borders obtained in the Invariant Border Experiment. For this evaluation, rather than training novel output layers on color bands that are evenly spaced, we train output layers with bands that are placed strategically within the category bounds. In 9 steps we train the output layer on bands placed within each border that are shifted from the left category border, to the right category border. Again, the classification is evaluated by testing the network on 60 samples for each point on the hue spectrum. To allow for detailed evaluation of the results, including noise, the classification results are plotted in *Appendix 9—figure 2*. Observing the results, we again see that some borders (notably between cyan and blue, as well as between blue and purple) are less invariant than other borders (notably the border between yellow and green). We also observe that in many some cases where a training band falls strongly on one side of the category, rather than borders shifting, the classification on the other side of the category can become noisier. While this leaves open the possibility that the concept of a color category is not as strong in the CNN as in humans, it is important to note that the network is not constrained to using color per se to perform the task and it can easily deviate to a different distinction. Note for instance that in the second row of *Appendix 9—figure 2*, the two orange patches are high in luminance (in terms of mean RGB).

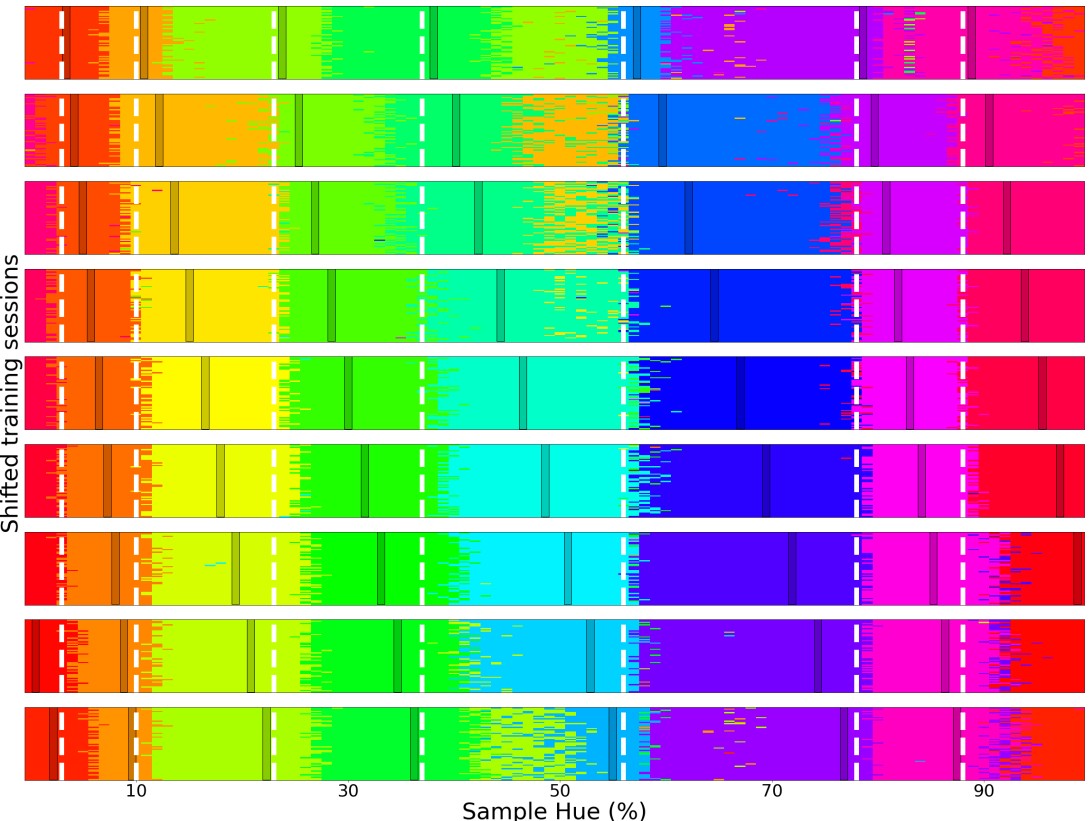

**Appendix 9—figure 2.** Each row displays the evaluation of a separate output layer trained on a set of color bands. These training bands are represented by darkened opaque vertical bands. Each layer is trained on 7 training bands, each falling within one of the categories found in the Invariant Border Experiment. After training, the output layer is evaluated by presenting it with 60 samples of each color on the hue spectrum (divided in 100 steps). Each pixel represents one classification sample and is colored for the training color of the class it is assigned to. To allow for comparing the discontinuities in each row with respect to the training bands and the category bounds from the original Invariant Border experiment, simultaneously, the latter have been added in the form of white vertical segmented lines.

