## [Editor Report]

This paper addresses the long-standing problem of color categorization and the forces that bring it about, which can be potentially interesting to researchers in cognition, visual neuroscience, society, and culture. In particular, the authors show that as a "model organism", a Convolutional Neural Network (CNN) trained with the human-labelled image dataset ImageNet for object recognition can represent color categories. The finding reveals important features of deep neural networks in color processing and can also guide future theoretical and empirical work in high-level color vision.

---

## [Decision Letter]

**Decision letter after peer review:**

Thank you for submitting your article "Color categories emerge in object learning" for consideration by *eLife*. Your article has been reviewed by 3 peer reviewers, and the evaluation has been overseen by a Reviewing Editor and Tirin Moore as the Senior Editor. The following individual involved in the review of your submission has agreed to reveal their identity: Bevil Conway (Reviewer #1).

Essential revisions:

Below is a list of revisions that we think are essential:

1) The authors need to experiment with other network architectures and different image datasets to show WHY and under WHAT circumstances color categorization may emerge or not emerge from a pre-trained network. Each of the three reviewers has offered some specific suggestions on how to do this. Please see their comments for details.

2) The authors need to show how their modeling results agree with existing psychophysical or physiological data of humans and animals. Otherwise, the biological relevance of the work might be limited. Please see Reviewer 3's comments for details.

3) Regarding the theoretical implications of their work, the authors are recommended to provide a more thorough discussion of the relationship of their findings with previous theories for color categorization and weaken claims such as "categories can emerge independent of language development". Please see Reviewers 1's and 3's comments.

*Reviewer #2 (Recommendations for the authors):*

Here are some of my suggestions:

They already apply k-means analysis on the image's pixels to show that there exist a few color clusters. But the clusters only match the revealed color category in other experiments partially. So they can also apply the similar k-means analysis of the units' responses in different layers of CNN to see how the color categorical information develops from layer to layer.

If the color categorical information emerges from the usefulness of object recognition, it might be interesting to see similar experiments with two other kinds of CNN as control:

1) Untrained CNN 2) CNN trained with images of which color information is not useful for object recognition. Recently, a few studies suggest that the untrained CNN can do face detection tasks (Baek et al., 2021) or do the enumeration task (Kim et al., 2021). Whether the color categorical perception can emerge from an untrained CNN will be very interesting. Also, if we really want to nail down the relationship between color categorical perception and object recognition, they might want to remove the relation between the color information and the object information. If a CNN is trained with these kinds of stimuli, can the similar color category emerge?

The paper lacks a direct comparison between the CNN and human subjects. The revealed color categories in CNN in the paper seem to be very reliable. Would these categories be similar to those observed in previous human behavior studies? If they were the same, that would strengthen the claim of the paper; if they were not, what are the possible reasons that would explain the difference?

*Reviewer #3 (Recommendations for the authors):*

The paper at this stage is a modeling paper without any human psychophysics. What we learned is behavior of a CNN. To my understanding, the only connection to humans is that the image dataset is labeled by humans. There are no comparisons to human psychophysics. To support the connection between the model results and human color categorization, especially the role of language and development in animal and human color categorization, the authors need to do more perceptual experiments or model previous empirical results from human vision. Otherwise, I suggest the authors remove some of these discussions and focus on the findings that color categorization might emerge through learning to classify objects. Perhaps, to strengthen the paper, an alternative model should be proposed (e.g. network trained NOT on a broad object dataset might NOT represent color categories) or results from an alternative CNN architecture other than ResNET.

---

## [Author Response]

Essential revisions:Below is a list of revisions that we think are essential:

We were very grateful for the thorough and constructive reviews.

1) The authors need to experiment with other network architectures and different image datasets to show WHY and under WHAT circumstances color categorization may emerge or not emerge from a pre-trained network. Each of the three reviewers has offered some specific suggestions on how to do this. Please see their comments for details.

We followed this advice and added a thorough analysis of different tasks, input images and network architectures. Importantly, training the network with a different task, results in a very different representation of color. Invariant color categories do not emerge when the network simply distinguishes natural scenes from man-made scenes, or when random hue shifts are applied to the input images. In repeating the object-task over different architectures we found that color categorization was prevalent in all. Thus, our results are not incidental, but a stable feature of object-recognition DNNs interacting with natural input data sets.

2) The authors need to show how their modeling results agree with existing psychophysical or physiological data of humans and animals. Otherwise, the biological relevance of the work might be limited. Please see Reviewer 3's comments for details.

We do agree with the reviewers that the lack of comparable psychophysical results was a weak point of our manuscript. We remedied this by devising a new psychophysical paradigm that is analogous to the network color classification task. Over 12,000 data points from 10 observers were collected and we find very similar results to those obtained with for the ResNet.

3) Regarding the theoretical implications of their work, the authors are recommended to provide a more thorough discussion of the relationship of their findings with previous theories for color categorization and weaken claims such as "categories can emerge independent of language development". Please see Reviewers 1's and 3's comments.

We agree that our claims were at times a bit strong. We revised the manuscript throughout and made sure not to overreach with our conclusions.

Reviewer #2 (Recommendations for the authors):Here are some of my suggestions:They already apply k-means analysis on the image's pixels to show that there exist a few color clusters. But the clusters only match the revealed color category in other experiments partially. So they can also apply the similar k-means analysis of the units' responses in different layers of CNN to see how the color categorical information develops from layer to layer.

We previously intended to leave this analysis for a future study. However, the reviewer’s question signaled the need to strengthen our conclusions. We have now included an analysis of how the color representation evolves over the layers throughout the network (see Appendix 6). We find that the color representation of the initial layer already strongly reflects the categorical structure. This could arise as a consequence of the input distribution, or could be due to the back-propagation of important image features to the early layers.

If the color categorical information emerges from the usefulness of object recognition, it might be interesting to see similar experiments with two other kinds of CNN as control:1) Untrained CNN 2) CNN trained with images of which color information is not useful for object recognition. Recently, a few studies suggest that the untrained CNN can do face detection tasks (Baek et al., 2021) or do the enumeration task (Kim et al., 2021). Whether the color categorical perception can emerge from an untrained CNN will be very interesting. Also, if we really want to nail down the relationship between color categorical perception and object recognition, they might want to remove the relation between the color information and the object information. If a CNN is trained with these kinds of stimuli, can the similar color category emerge?

We have now analyzed both an untrained network (random weight network) to look at the behavior inherent to the CNN and a network trained on the same images, but with a random hue-shift applied to each input image (random hue network).

Using a randomly initialized network we find a rather noisy output with borders jumping considerably as the training bands are shifted. However, through this noise we do see a general tendency to form invariant borders, indicating that the notion to form categories may be inherent to the structure of the network.

When we decouple colors from the original by randomly shifting the hues for each input, we see that representation no longer follows the hue spectrum and clear borders can no longer be identified in the representation indicating that the distribution in the input set is crucial in determining the border locations.

The details can be found in the Appendix 7.

The paper lacks a direct comparison between the CNN and human subjects. The revealed color categories in CNN in the paper seem to be very reliable. Would these categories be similar to those observed in previous human behavior studies? If they were the same, that would strengthen the claim of the paper; if they were not, what are the possible reasons that would explain the difference?

We have now run a human psychophysics experiment closely modelled on the match-to-sample task the CNN performs. The results are very similar, as we see that categories are narrower in the red to yellow part of the spectrum and border transitions are not entirely invariant. Please see the novel section “Human psychophysics” for the details (page 8).

Reviewer #3 (Recommendations for the authors):The paper at this stage is a modeling paper without any human psychophysics. What we learned is behavior of a CNN. To my understanding, the only connection to humans is that the image dataset is labeled by humans. There are no comparisons to human psychophysics. To support the connection between the model results and human color categorization, especially the role of language and development in animal and human color categorization, the authors need to do more perceptual experiments or model previous empirical results from human vision. Otherwise, I suggest the authors remove some of these discussions and focus on the findings that color categorization might emerge through learning to classify objects.

We understand the lack of human data considerably limits the strength of our conclusions; we have now added a human psychophysical experiment to be able to compare the results from the CNN to human data. Please see “Human Psychophysics” on page 7.

Perhaps, to strengthen the paper, an alternative model should be proposed (e.g. network trained NOT on a broad object dataset might NOT represent color categories) or results from an alternative CNN architecture other than ResNET.

To the reviewer’s points, we have now added three new analyses.

– Firstly, we have trained and analyzed a network on distinguishing natural vs artificial images using the same dataset. With this, we find that the color representation the network creates to perform classification on does not entail the same color representation. (see the task analysis on page 7)

– Secondly, we have trained a network on the same dataset, but now with random hue shifts applied to each input as they are fed to the network. We again find a very different color representation. (see Appendix 7)

– Thirdly, we have now introduced a section in the Appendix that evaluates several different CNNs and find strong similarities across the color representations of the networks (see Appendix 8).

While point 1 and 2 slightly deviate from the reviewer’s suggestion, we think it does answer the question she/he is after. We only deviate slightly, as there is great overlap with points by other reviewers and we have tried to find a unified approach to cover similar questions to avoid adding too many analyses focusing on the same question.